# Nearly Optimal Bounds for Cyclic Forgetting

**Halyun Jeong**
University of California Los Angeles
hajeong@math.ucla.edu

**Mark Kong**
University of California Los Angeles
markkong@ucla.edu

**Deanna Needell**
University of California Los Angeles
deanna@math.ucla.edu

**William Swartworth**
Carnegie Mellon University
wswartwo@andrew.cmu.edu

**Rachel Ward**
University of Texas at Austin
rward@math.utexas.edu

## Abstract

We provide theoretical bounds on the forgetting quantity in the continual learning setting for linear tasks, where each round of learning corresponds to projecting onto a linear subspace. For a cyclic task ordering on $T$ tasks repeated $m$ times each, we prove the best known upper bound of $O(T^2/m)$ on the forgetting. Notably, our bound holds uniformly over all choices of tasks and is independent of the ambient dimension. Our main technical contribution is a characterization of the union of all numerical ranges of products of $T$ (real or complex) projections as a sinusoidal spiral, which may be of independent interest.

## 1 Introduction

While machine learning often focuses on learning from a static dataset, one is often interested in acquiring data on the fly and adapting to the present context. This is known throughout the literature as "continual learning" or "lifelong learning" [Wan+23; CL18; Par+19; TM95].

One challenge of continual learning is "catastrophic forgetting" [Had+20; VT19; Kem+18]: A model may learn useful information in context $A$ but then lose much of this knowledge when adapting to context $B$. If context $A$ is not representative of future contexts, then this is a correct adjustment for distribution shift. However, if contexts similar to $A$ arise repeatedly, this may be undesirable..

Building on prior work, we aim to better understand the catastrophic forgetting phenomenon when faced with a series of linear regression problems that are revisited in cyclic order.

In machine learning, many data sets display cyclic or periodic patterns. These patterns often arise from factors such as the "day of the week effect" [QR22] observed in financial or search engine data [Yan+22]. Additionally, the methods of cyclic alternating projections by Von Neumann [Von49] and Halperin [Hal62] are well-studied methods for solving linear systems [Kar37]. Equivalently, one can think of this as studying residual bounds for cyclic block Kaczmarz-type algorithms. Our work can be thought of as studying the worst-case forgetting of these popular methods. [Evr+22] mentions this perspective, but leaves open the problem of obtaining tight convergence bounds. In particular, the dimension dependence on the ambient data in the bounds of [Evr+22] could be problematic since high-dimensional data is so ubiquitous in machine learning. In contrast, our new bound for the worst-case forgetting does not scale at all with the dimension of the ambient data.

37th Conference on Neural Information Processing Systems (NeurIPS 2023).

## 1.1 Problem Setup

The learning algorithm we consider fits a linear map to a set of data points of the form $(\vec{x}, y)$ with $\vec{x} \in \mathbb{R}^d, y \in \mathbb{R}$.

Our data is split into "tasks", which are lists of elements from our dataset (with repeats allowed). These tasks are fed to our learning algorithm one at a time. Our algorithm updates its prediction after being given each task.

We assemble the $r_t$ inputs of the $t$th task into a matrix $X_t \in \mathbb{R}^{r_t, d}$, where each row is an input vector $\vec{x}^T$, and their corresponding outputs into a column vector $\vec{y}_t \in \mathbb{R}^{r_t}$. We aim to to find a column vector $\vec{w} \in \mathbb{R}^d$ such that $X_t \vec{w} = \vec{y}_t$ for all $t$.

The learning algorithm we consider applies a block Kaczmarz update at each step. Explicitly, we initialize $\vec{w}_0 \in \mathbb{R}^d$ anywhere, and for $t \in \mathbb{Z}_{\geq 1}$ we set $\vec{w}_t$ to be the projection of $\vec{w}_{t-1}$ onto the solution space $P_t$ for the $t$th regression problem. That is,

$$\vec{w}_{t+1} = \vec{w}_t + X_{t+1}^+(\vec{y}_{t+1} - X_{t+1}\vec{w}_t)$$

where $+$ denotes the Moore-Penrose pseudoinverse.

Geometrically, the set of vectors $\vec{w} \in \mathbb{R}^d$ such that the map $\mathbb{R}^d \to \mathbb{R} \colon \vec{x} \mapsto \vec{x}^T \vec{w}$ agrees with all the data in task $t$ is an affine subspace $P_t$. This algorithm obtains $w_{t+1}$ by projecting $w_t$ onto $P_{t+1}$. The set of vectors in $\mathbb{R}^d$ satisfying all tasks is $P := \bigcap_t P_t$. The algorithm never leaves the maximal affine hyperplane through $\vec{w}_0$ perpendicular to $P$, so if the $w_t$ converge to a solution then it converges to the solution closest to $w_0$..

It is well-known that running gradient descent to interpolation with quadratic loss on task $t$ starting from $w_t$ produces $w_{t+1}$.

Typically, projecting onto the solution space for task $t$ moves $\vec{w}_{i-1}$ off the solution space for task $t-1$. We wish to understand the amount of information "forgotten" over the course of training. Following [Evr+22], we define the **forgetting** for a sequence of tasks and initialization $S$ at iteration $n$ as

$$F_S(n) := \frac{1}{n} \sum_{t=1}^n \|X_t \vec{w}_n - \vec{y}_t\|^2 .$$

That is, $F_S(n)$ is the average loss of $w_n$ on all previous tasks (including task $n$), where each task is weighted equally regardless of how many data points it has. As mentioned in the introduction, large forgetting is not always undesirable, but "forgetting" in humans is not always undesirable either, so the name "forgetting" not imply an inappropriately negative connotation.

In this work, we focus on cyclic task orderings. That is, we have $T$ tasks $(X_1, \vec{y}_1), \ldots, (X_T, \vec{y}_T)$ we visit cyclically, i.e., $X_t = X_{t \mod T+1}$ and $\vec{y}_t = \vec{y}_{t \mod T+1}$ for all $t$. Then the sequence of iterates converges to a cycle of length $T$ in the sense that, for any $t \in \mathbb{Z}_{\geq 0}$, $w_t, w_{t+T}, w_{t+2T}, \ldots$ converges, either linearly or after a finite number of iterations; We justify this in the supplementary material.

In particular, if there is at least one simultaneous solution to all tasks, then the limit must be a solution; We justify a generalization of this in the supplmentary material, so we get linear convergence to a solution which implies linear convergence of forgetting.

However, this does not guarantee that, for any collection of $T$ tasks in $m$ cycles, the forgetting can be bounded above by $O(c^{-m})$ for some constant $c$, even after normalizing the tasks in the way we describe later: The choice of tasks that give the largest forgetting for one value of $m$ may not be the tasks that give the largest forgetting for another value of $m$. Here we ask whether we can obtain a worst-case bound on $F_S(mT)$ that only depends on the number of tasks $T$ and the number of cycles $m$.

We focus on bounding the forgetting after a whole number of cycles; The forgetting in the middle of a cycle is a weighted average of the losses at each task, and therefore can be treated in the same way as getting a bound without normalizing the tasks. For a quick bound, the weights are close to 1, so we may rescale the tasks to get all coefficients to be 1 or less without increasing our bound much. The forgetting for $T$ tasks after $m$ cycles is

$$\frac{1}{mT} \sum_{t=1}^{mT} \|X_t \vec{w}_{mT} - \vec{y}_t\|^2 = \frac{1}{T} \sum_{t=1}^{T} \|X_t \vec{w}_{mT} - \vec{y}_t\|^2 . \tag{1}$$

as each of the $T$ tasks is visited equally often.

This can be made arbitrarily large by scaling the data points, so we describe normalizations in the next section. Given these normalizations, [Evr+22] showed that such a worst-case bound is possible, but the bounds they provided are not tight. Here, we improve on the forgetting bounds from [Evr+22] and, as a byproduct, show the union of the numerical ranges of all products of $T$ orthogonal (real or complex) projections forms a sinusoidal spiral.

### 1.1.1 Normalizations

Following [Evr+22], we restrict to the setting where $P$ is nonempty (i.e., the union of all tasks is consistent with a single linear function) and let $\vec{w}_* \in P$. So every data point $(\vec{x}, y)$ satisfies $\vec{x}^T \vec{w}_* = y$, and the planes $P_i$ we project onto all pass through $\vec{w}_*$.

Unlike [Evr+22], we translate the setup so $w_* = 0$. This modifies each data point $(\vec{x}, y)$ by setting $y = 0$ but preserving $\vec{x}$. This translates the $P_i$ to pass through 0, so they are subspaces, and $P_i$ is the kernel of $X_i$, and the loss of task $t$ at $\vec{w}$ is now $\|X_t \vec{w}\|^2$, turning forgetting into.

$$F_S(n) = \frac{1}{n} \sum_{t=1}^{n} \|X_t \vec{w}\|^2 = \frac{1}{n} \left\| \left( \underset{t=1}{\overset{n}{\times}} X_t \right) (\vec{w}) \right\|^2$$

where the product is the categorical product. Thus, for a fixed collection of tasks $X_1, X_2, \ldots, X_n$, abusing notation to let $P_t$ denote the projection onto the plane $P_t$, the largest forgetting over all initializations $\vec{w}_0$ is $\dfrac{\left\| \left( \underset{t=1}{\overset{n}{\times}} X_t \right) \left( \underset{t=1}{\overset{n}{\prod}} P_t \right) \right\|^2}{n}$.

Rescaling $\vec{w}_0$ rescales all iterates, and hence the forgetting, by the same amount. So we normalize $\|\vec{w}_0\| = 1$. Similarly, rescaling all tasks $X_t$ by the same scalar scales each term in the forgetting by that factor, so we normalize so that $\max_t \|X_t\| = 1$.

As $\|\vec{w}_t\|$ decreases in $t$, this implies the loss of $\vec{w}_n$ at task $t$ is $\|X_t \vec{w}_n\|^2 \leq 1$, and hence the forgetting is at most 1.

In the cyclic setting, for fixed $T$ and $m$, these normalizations make the domain of optimization (the space of $T$-tuples of tasks with $\max_t \|X_t\| \leq 1$ and initializations $\vec{w}_0 \in S^{d-1}$) compact, so forgetting attains a maximum, which is what we bound in this paper.

In the supplementary, we describe a tighter normalization that gives the same forgetting bounds.

## 1.2 Observations

In this section, we describe some observations that help us get our bound.

As noted in the previous section, using our normalization, there is a trivial bound on the forgetting of 1.

The data matrices $X_t$ only affect the iterates $\vec{w}_i$ via their rowspan (i.e., the span of the data points in the task), so we may replace $X_i$ by any other data matrix with the same rowspan, or equivalently the same kernel, without affecting the iterates subject to our normalization constraint. As $\|X_i\| \leq 1$, it is a contraction, so the norm of the image of any $\vec{w}$ is at most its distance to the kernel, and this is realized when $X_i$ is a projection away from its kernel. Therefore, the worst-case bound is not affected by restricting $X_i$ to be orthogonal projections with kernel $P_i$.

With this restriction, for any $\vec{w}$, $\|X_i \vec{w}\|$ is the distance from $\vec{w}$ to $P_i$. Therefore, the forgetting can be interpreted as the average distance to each $P_i$ over all previous tasks (weighted equally).

## 1.3 Prior Work

**Bounds on forgetting.** [Evr+22] considered forgetting bounds both for random task orderings and cyclic task orderings. However their bound for cyclic task orderings is not tight. After $m$ cycles of $T$

tasks, they show an upper bound of $\min\left\{\frac{T^2}{\sqrt{mT}}, \frac{T^2 d}{2mT}\right\}$ on the forgetting, leaving a gap with their lower bound of $\Omega(\frac{T^2}{mT})$

A key step of their proof is bounding $\left\|A^m\vec{u}\right\|^2 - \left\|A^{m+1}\vec{u}\right\|^2$ where $A$ is a product of $T$ projections and $u$ is a unit vector. Their dimension-independent bound gives a $T^2/\sqrt{mT}$ dependence on the forgetting after $m$ cycles. Their other bound improves the $m$ dependence, but only uses that $A$ is a contraction. This forces a linear dependence on the dimension $d$, as this is possible if the supremum is taken over all contractions. Indeed, for any $m$ a multiple of $d$, one can have

$$\left\|A^m\vec{u}\right\|^2 - \left\|A^{m+1}\vec{u}\right\|^2 \geq \Omega(d/m)$$

by choosing $A$ to be an operator that maps

$$\vec{e}_1 \mapsto \vec{e}_2 \mapsto \ldots \mapsto \vec{e}_d \mapsto (1 - \frac{d}{m})\vec{e}_1,$$

and taking $\vec{u} = \vec{e}_1$, where $\vec{e}_i$ denotes the $i$th standard unit vector.

[RZ23] bound $\left\|A^m\vec{u} - A^{m+1}\vec{u}\right\|^2$ for $A$ a product of projections. However they are primarily interested in the asymptotics with respect to $m$. They supply two proofs which rely on either the bound in [BS16] or in [Cro08], and hence do not avoid the exponential dependence in $T$.

[Lin+23] study forgetting from a different perspective. They assume Gaussian i.i.d. tasks ( called features in their paper) at each iteration and explicitly compute the expected forgetting and generalization error. We analyze the worst-case forgetting without any assumptions on the distribution of tasks, so our work is quite different from [Lin+23].

As mentioned in [Evr+22], the iterate $w_t$ can be equivalently obtained by running the (block) Kaczmarz method [NT14] with the cyclic row selection rule for solving a consistent system $y_t = X_t w, \forall t \in [T]$. However, most id=MK]of the convergence analyses of Kaczmarz-type methods [Elf80; NT14; SV09] bound the distance from the least-norm solution, whereas in our setting we interested in the residual errors. Moreover such analyses depend on the condition number of the system, so are unlikely to adapt to our setting, since our bounds hold for all choices of tasks (i.e. blocks), with no condition number dependence.

**Bounds on the numerical range.** Lemma 5.1 in [Cro08] gives a bound on the numerical range of any linear transformation $A$ expressible as a product of $T$ projections. This, combined with other results in that same work, yields a bound on $\left\|A^m\vec{u} - A^{m+1}\vec{u}\right\|^2$ . When combined with the results of [Evr+22], the result is a bound of the form $O(f(T)/m)$ on the forgetting. However, the function $f(T)$ turns out to be exponential when applying the results of [Cro08] directly. This is due to Lemma 5.1 of [Cro08] being loose. There they show that the numerical range of $A$ is contained in a region of $\mathbb{C}$ with a boundary point at one where the tangent lines at 1 have exponential slope. In this paper, we give the smallest region that is guaranteed to contain the numerical range of $A$ given only $T$.

[BS16] also gives a bound on the numerical range of a product of projections. Specifically they show that the numerical range is contained in a Stoltz domain with the angle at 1 depending on a quantity called the Friedrich's number of the (subspaces associated to the) projections. They then combine their bound with that of [Cro08] to obtain somewhat tighter control over the numerical range. However the Friedrich's number for $T$ subspaces can be made arbitrarily close to 1 so for our purposes their bound will not yield better results than the bounds in [Cro08].

## 1.4   Results

We give a nearly optimal bound on the forgetting for cyclic task orderings, removing the dimension dependence of [Evr+22]and avoiding the exponential dependence on $T$ that would arise from applying results in [Cro07].

**Theorem 1.**

$$\sup_S F_S(mT) \leq O\left(\frac{T^2}{m}\right).$$

*where $S$ ranges over all (task, initialization) pairs where the tasks are given by matrices $X_1, X_2, \ldots, X_T$ with $\max_t \|X_t\| \leq 1$ and $\vec{w}_0 \in \mathbb{R}^d$ with $|\vec{w}_0| = 1$.*

To prove this, [Evr+22] show that, if $A$ denotes the linear map given by one cycle, letting $\Delta_m(\vec{u}) := \|A^m \vec{u}\| - \|A^{m+1} \vec{u}\|$,

$$\sup_S F_S(mT) \leq \frac{T-1}{2} \cdot \max_{\vec{u}:\|u\|=1} \Delta_m(\vec{u}) \leq \frac{T-1}{2} \|A^m(1-A)\|.$$

We provide a geometric interpretation of their proof in the supplementary material. [CP17] showed that there exists a constant $Q \leq 1 + \sqrt{2}$ such that

$$\|p(A)\| \leq Q \sup_{z \in W(A)} |p(z)|$$

for all polynomials $p$. Our approach is to determine $\bigcup_A W(A)$ where $A$ ranges over all products of $k$ complex orthogonal projections. As this includes the class of products of $k$ real orthogonal projections, combining these inequalities gives our bound.

We will show in Section 3 that the problem of bounding the numerical range of the product of $k-1$ complex orthogonal projections is equivalent to understanding the range of the product

$$P(\vec{v}_0, \ldots, \vec{v}_{k-1}) := \langle \vec{v}_0, \vec{v}_1 \rangle \langle \vec{v}_1, \vec{v}_2 \rangle \ldots \langle \vec{v}_{k-2}, \vec{v}_{k-1} \rangle \langle \vec{v}_{k-1}, \vec{v}_0 \rangle, \tag{2}$$

where the $\vec{v}_i$ range over complex vectors in $\mathbb{C}^d$ of norm 1. We give a complete description of the range of $P$ as the *filled sinusoidal spiral*

$$re^{i\theta} \; : \; \cos(\pi/k)r^{-1/k} \geq \cos(\frac{\theta - \pi}{k}).$$

This shape has a simple geometric description. Let $\mathcal{P}_k$ be the convex hull of the $k$th roots of unity. Then the sinusoidal spiral above is the image of $\mathcal{P}_k$ under the map $z \mapsto z^k$.

The main challenge in computing $P$'s range $\Gamma_k$ is in computing its boundary since a topological argument (given in the supplementary material) shows that $\Gamma_k$ is simply connected. To compute the boundary, we first observe that it suffices to consider input vectors in $\mathbb{C}^2$. To see this, $P$ maps a sequence of vectors to the cyclic product of pairwise inner products, so if any of the input vectors is not coplanar with its neighbors, then projecting it onto the plane spanned by its neighbors and scaling it to have unit norm increases the magnitude of the corresponding factors. So it suffices to consider the case when all vectors are coplanar, which is equivalent to considering input vectors in $\mathbb{C}^2$. In this case, $P$ is a smooth map from the (real) manifold $(S^1)^n \subset \mathbb{C}^n$ to the (real) manifold $\mathbb{C}$, so any input that gets sent to a boundary point of $\Gamma_k$ has singular Jacobian. That is, the directional derivative in all directions tangent to the domain must be parallel. It turns out that these algebraic conditions can be manipulated to characterize all critical points and critical values of $P$, and the computation is made simpler by using quaternions.

As a side effect of our proof, we also fully describe the tuples $\vec{v}_0, \ldots, \vec{v}_{k-1}$ that map to boundary points of the image of $P$ and show that they naturally correspond (up to multiplying each $\vec{v}_t$ by a complex unit $\phi_t$) to certain quaternionic roots of unity.

Using this result we obtain the following bound on the increments between consecutive powers of $A$ when $A$ is a product of projections. We believe this result may be independently useful.

**Lemma 2.** *For any linear map* $A \colon \mathbb{C}^n \to \mathbb{C}^n$ *expressible as a product of $k$ complex projections,*

$$\|(I - A)A^m\| \leq (C + o_{k, m \to \infty}(1)) \frac{k}{m},$$

*where $C$ is an absolute constant which can be taken to be* $0.4$.

This lemma, with the inequality of [Evr+22], immediately implies Theorem 1.

In particular, for fixed $T$, we obtain an asymptotically optimal bound in terms of iteration count $m$, *independent of dimension*. In contrast, the dimension-independent bound given in [Evr+22] was sub-optimal by a factor of $\sqrt{m}$.

Finally, . So our results can be interpreted as giving a bound for this setting as well.

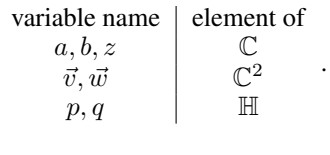

| variable name | element of |
|---|---|
| $a, b, z$ | $\mathbb{C}$ |
| $\vec{v}, \vec{w}$ | $\mathbb{C}^2$ |
| $p, q$ | $\mathbb{H}$ |

.

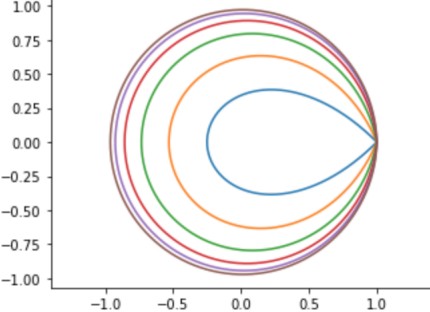

Figure 1: The boundaries of $\Gamma_m$ for $m = 2^2, 2^3, \ldots, 2^7$.

## 1.5 Notation for proof

We use $\mathbb{C}$ and $\mathbb{H}$ to denote the complex numbers and the quaternions. While the previous sections considered real vector spaces, our proof works by obtaining a bound over all complex projections. So for sections 2 and 3, all vector spaces will be over $\mathbb{C}$. In these sections, we use the following guidelines for variable names:

Our complex inner products are always conjugate linear in the first argument, so $\langle \vec{v}, \vec{w} \rangle = \vec{v}^* \vec{w}$ where $v^*$ denotes the conjugate transpose of $\vec{v}$. The argument of a complex number is taken to be in $(-\pi, \pi]$. The numerical range of a matrix $A \in \mathbb{C}^{d \times d}$ is defined by

$$W(A) = \{\vec{z}^* A \vec{z} | \vec{z} \in \mathbb{C}^d, \|\vec{z}\| = 1\}.$$

## 2 Bounding the range of $P$

In this section, we fully describe the range of $P$ given by (2) and obtain a full description of the inputs that map to boundary points. We will use the following notation throughout the next argument.

**Definition 3.** *For $(a, b) \in \mathbb{C}^2$, we define $(a, b)^\perp := (\bar{b}, -\bar{a})$.*

The following identity is a straightforward consequence of the definition.

**Proposition 4.** *For $\vec{v}, \vec{w} \in \mathbb{C}^2$, $\langle \vec{v}, \vec{w}^\perp \rangle = -\overline{\langle \vec{v}^\perp, \vec{w} \rangle}$.*

Our first theorem shows that the range of $P$ is a filled sinusoidal spiral. This result is of mathematical interest on its own, and will also be applied to cyclic forgetting bounds in the next section.

We begin with a lemma characterizing the critical points of $P$ treated as a smooth map $\mathbb{R}^{2k} \to \mathbb{R}^2$. We defer the proof to the supplementary material.

**Lemma 5.** *For any critical value $c \in \mathbb{C}$ of $P$, there exists a sequence of vectors $(a_0, b_0) = (1, 0), (a_1, b_1), \ldots, (a_{k-1}, b_{k-1})$ such that $P((a_m, b_m)_{m=0}^{k-1}) = c$. Furthermore, there exist $\alpha, \beta \in \mathbb{C}$ such that*

*1. $\langle (a_m, b_m), (a_{m+1}, b_{m+1}) \rangle = \alpha$ for all $m$*

*2. $\langle (a_m, b_m)^\perp, (a_{m+1}, b_{m+1}) \rangle = \beta \omega^{m-1}$ for all $m$*

*(where indices are treated mod $k$). Furthermore, any sequence $((a_m, b_m)_{m=0}^{k-1})$ attaining $P(((a_m, b_m)_{m=0}^{k-1})) = c$ can be obtained by taking a sequence satisfying the above conditions and multiplying each vector by a complex unit.*

Given this characterization of $P$'s critical points, our next result computes the boundary points of $P$'s image, from which we obtain the entire image of $P$ as a consequence.

**Theorem 6.** *Let* $P(\vec{v}_0, \ldots, \vec{v}_{k-1}) = \langle \vec{v}_0, \vec{v}_1 \rangle \langle \vec{v}_1, \vec{v}_2 \rangle \ldots \langle \vec{v}_{k-2}, \vec{v}_{k-1} \rangle \langle \vec{v}_{k-1}, \vec{v}_0 \rangle$ *and let* $\Gamma_k$ *be the range of* $P$ *as each* $\vec{v}_i$ *ranges over unit vectors in* $\mathbb{C}^d$. *Then*

$$\Gamma_k = \left\{ re^{i\theta} \mid \cos\left(\frac{\pi}{k}\right) r^{-1/k} \geq \cos\left(\frac{\theta - \pi}{k}\right), r \geq 0, \theta \in [0, 2\pi] \right\},$$

*which is a filled sinusoidal spiral.*

*Proof.* We show this result for vectors in $\mathbb{C}^2$. In the supplementary material, we show that increasing the dimension does not affect the range, so considering vectors in $\mathbb{C}^2$ is sufficient. The result is trivial for $k \leq 2$, so we will assume $k \geq 3$.

The domain of $P$ is compact, so $\Gamma_k$ is also compact. $P$ is also smooth (as a function $\mathbb{R}^4 \to \mathbb{R}^2$), so its boundary consists of critical values. We will give a complete description of the critical values and critical points and show that the outermost critical values form a sinusoidal spiral, which is enough to show that $\Gamma_k$ is contained in the claimed region. We defer showing that $\Gamma_k$ contains the entire filled spiral to the supplementary material.

Fix a critical value $c$ of $P$ and let $a_i, b_i$ be as in Lemma 5 above. As $(a_m, b_m)$ and $(a_m, b_m)^\perp$ are orthogonal, they form a basis for $\mathbb{C}^2$, so letting $\alpha = a_0, \beta = b_0$, we get a recurrence relation

$$(a_1, b_1) = \alpha(1, 0) + \beta(1, 0)^\perp$$

$$(a_2, b_2) = \alpha(a_1, b_1) + \beta\omega(a_1, b_1)^\perp$$

$$(a_3, b_3) = \alpha(a_2, b_2) + \beta\omega^2(a_2, b_2)^\perp$$

$$\vdots$$

$$(a_{k-1}, b_{k-1}) = \alpha(a_{k-2}, b_{k-2}) + \beta\omega^{k-1}(a_{k-2}, b_{k-2})^\perp$$

$$(1, 0) = (a_0, b_0) = \alpha(a_{k-1}, b_{k-1}) + \beta\omega^k(a_{k-1}, b_{k-1})^\perp.$$

Representing each vector $(a, b) \in \mathbb{C}^2$ as a quaternion via $(a, b) \mapsto a + bj$, it follows from direct computation that $(a, b)^\perp \mapsto j(a + bj)$ and, for $\phi \in \mathbb{C}$, $\phi(a, b) \mapsto \phi(a + bj) = (a + bj)\bar{\phi}$. In particular, multiplication on the left by a complex number is unambiguous.

Our recurrence relation can be expressed as

$$a_1 + b_1 j = (\alpha + \beta j)(1 + 0j)$$

$$a_2 + b_2 j = (\alpha + \beta\omega j)(a_1 + b_1 j)$$

$$a_3 + b_3 j = (\alpha + \beta\omega^2 j)(a_2 + b_2 j)$$

$$\vdots$$

$$a_{k-1} + b_{k-1} j = (\alpha + \beta\omega^{k-2} j)(a_{k-2} + b_{k-2} j)$$

$$1 + 0j = (\alpha + \beta\omega^k)(a_{k-1} + b_{k-1} j).$$

In other words, the vectors must be the partial products

$$(a_m, b_m) = (\alpha + \beta\omega^{m-1} j)(\alpha + \beta\omega^{m-1} j) \ldots (\alpha + \beta\omega^0 j)$$

for $m = 0, \ldots, k - 1$ (which is the empty product of $m = 0$), where $\alpha, \beta$ satisfy the equation

$$1 = (\alpha + \beta\omega^{k-1} j)(\alpha + \beta\omega^{k-2} j) \ldots (\alpha + \beta\omega^0 j).$$

We now show that any sequence of vectors $(a_0, b_0), \ldots, (a_{k-1}, b_{k-1})$ in a solution to the above system of equations can be obtained by slightly modifying a solution in the case $\omega = 1$ (possibly with a different value of $k$).

Let $\zeta_k$ be a (not necessarily primitive) complex $k$th root of unity. Consider the sequence

$$(a_0, b_0), (a_1\zeta_k, b_1\zeta_k), (a_2\zeta_k^2, b_2\zeta_k^2), \ldots, (a_{k-1}\zeta_k^{k-1}, b_{k-1}\zeta_k^{k-1}).$$

This also satisfies the two conditions in Lemma 5, so we can run the same logic as above. The corresponding change in the resulting system is that this multiplies $\alpha, \beta$ by $\zeta_k$ and $\omega$ by $\zeta_k^2$, so by multiplying each vector by these complex units we may freely modify $\omega$ by $\zeta_k^2$ for any $k$th root of unity $\zeta_k$, or equivalently any power of $e^{2\pi i/k}$.

If $k$ is odd, this is enough to show that any nonzero critical value can be obtained by $\omega = 1$, and furthermore that any critical point can be expressed as $(a_0, b_0), (a_1\zeta_k, b_1\zeta_k), (a_2\zeta_k^2, b_2\zeta_k^2), \ldots, (a_{k-1}\zeta_k^{k-1}, b_{k-1}\zeta_k^{k-1})$ for some $k$th root of unity $\zeta_k$.

If $k$ is even, let $\zeta_{2k}$ be a $2k$th root of unity. Consider the sequence

$$(a_0, b_0), (a_1\zeta_{2k}, b_1\zeta_{2k}), (a_2\zeta_{2k}^2, b_2\zeta_{2k}^2), \ldots, (a_{k-1}\zeta_{2k}^{k-1}, b_{k-1}\zeta_{2k}^{k-1}),$$
$$(a_0\zeta_{2k}^k, b_0\zeta_{2k}^k), (a_1\zeta_{2k}^{k+1}, b_1\zeta_{2k}^{k+1}), \ldots, (a_{k-1}\zeta_{2k}^{2k-1}, b_{k-1}\zeta_{2k}^{2k-1}).$$

This satisfies the conditions in Lemma 5 for $2k$ in place of $k$. The corresponding values of $\alpha, \beta$ in the resulting system are the same as in the original multiplied by $\zeta_{2k}$, and the resulting value of $\omega$ is the same as the original multiplied by $\zeta_{2k}^2$. As $\zeta_{2k}^2$ can be chosen to be $\omega^{-1}$, we can reduce to the case

$$1 = (\zeta_{2k}(\alpha + \beta\omega^{2k-1}j))(\zeta_{2k}(\alpha + \beta\omega^{2k-2}j))\ldots(\zeta_{2k}(\alpha + \beta\omega^0 j))$$

where $\omega = 1$, or equivalently

$$1 = (\zeta_{2k}(\alpha + \beta j))^{2k}.$$

**Remark 7.** *These reductions can also be interpreted using quaternions, which we demonstrate in the supplementary material.*

For any integer $m$, the $m$th roots of unity in $\mathbb{H}$ are precisely those numbers whose real parts are the same as those of the real parts of the $m$th roots of unity in $\mathbb{C}$. So $(\alpha + \beta j)^m = 1$ if and only if $\alpha$ lies on one of the vertical diagonals of the $m$-gon whose vertices are the $m$th roots of unity (where the vertical diagonals are those with constant real part, including sides of the $n$-gon).

Therefore, the values of $\alpha$ in the phase-shifted sequences are vertical diagonals in the regular $k$-gon if $k$ is odd and every other vertical diagonal in the regular $2k$-gon if $k$ is even, where the vertices are the corresponding roots of unity. Inverting the phase shift gives that, regardless of whether $k$ is even or odd, the values of $\alpha$ in the solutions to

$$1 = (\alpha + \beta\omega^{k-1}j)(\alpha + \beta\omega^{k-2}j)\ldots(\alpha + \beta\omega^0 j)$$

form the diagonals (including the sides) of the $k$-gon whose vertices are the $k$th roots of unity.

As the inner product of consecutive vectors at such a sequence is $\alpha$, the value of $P$ at such a sequence is $\alpha^k$. Therefore the critical values of $f$ are then $k$th powers of these diagonals. Furthermore, when $k$ is odd the critical points are sequences of the form $(a_m, b_m) = \phi_m(\alpha + \beta j)^m$ where $\alpha + \beta j$ is a quaternionic $k$th root of unity and $\phi_m$ are units in $\mathbb{C}$; When $k$ is even, the critical points are sequences of the form $(a_m, b_m) = \phi_m(\alpha + \beta j)^m$ where $\alpha + \beta j$ are quaternionic $2k$th roots of unity and $\phi_m$ are units in $\mathbb{C}$,

By the maximum principle for holomorphic maps, the outer boundary of $\Gamma_k$ is obtained by taking the $k$th powers of the sides (where each side maps to the same curve). Using polar coordinates, the resulting curve is the sinusoidal spiral $\cos(\frac{\pi}{k})r^{-1/k} = \cos(\frac{\theta-\pi}{k})$. $\qquad \square$

The same proof shows that all other critical points other than those that $P$ sends to 0 arise from the other diagonals of the polygons and their corresponding critical values are their $k$th powers, which are also sinusoidal spirals. A computation shows that a sequence of vectors that $P$ sends to 0 is a critical point if and only if at least two of the inner products are 0.

## 3  Application to cyclic forgetting bounds

As an immediate application of Theorem 6, we can control the numerical range of a product of $k$ orthogonal projections.

**Corollary 8.** *Let $A = P_k P_{k-1} \ldots P_1$ be a product of $k$ orthogonal complex projections $\mathbb{C}^d \to \mathbb{C}^d$. Then $W(A) \subseteq \Gamma_{k+1}$.*

*Proof.* We first observe a general fact. Let $P\colon \mathbb{C}^d \to \mathbb{C}^d$ be an orthogonal projection and let $\vec{z} \in \mathbb{C}^d$. Let $\vec{q} = \frac{P\vec{z}}{\|P\vec{z}\|}$. Then

$$P\vec{z} = \vec{q}\langle\vec{q}, \vec{z}\rangle = \vec{q}\vec{q}^{\,*}\vec{z}.$$

Now let $\vec{z}$ be an arbitrary unit vector and let

$$\vec{q_i} = \frac{P_i P_{i-1} \ldots P_1 \vec{z}}{\|P_i P_{i-1} \ldots P_1 \vec{z}\|}.$$

Iterating the fact stated above,

$$(\vec{q_i}\vec{q_i}^{\,*}) \ldots (\vec{q_1}\vec{q_1}^{\,*})\vec{z} = P_i \ldots P_1\vec{z},$$

for all $i \in [k]$. Hence

$$\vec{z}^{\,*}A\vec{z} = \vec{z}^{\,*}(\vec{q_k}\vec{q_k}^{\,*})(\vec{q_{k-1}}\vec{q_{k-1}}^{\,*}) \ldots (\vec{q_1}\vec{q_1}^{\,*})\vec{z} = \langle\vec{z}, \vec{q_k}\rangle \langle\vec{q_k}, \vec{q_{k-1}}\rangle \ldots \langle\vec{q_2}, \vec{q_1}\rangle \langle\vec{q_1}, \vec{z}\rangle,$$

so $W(A) \subseteq \Gamma_{k+1}$. $\qquad\square$

In the supplementary material, we use our characterization of $P^{-1}(\partial\Gamma_{k+1})$ to characterize the pairs $(A, \vec{z})$ such that $\vec{z}^{\,*}A\vec{z} \in \partial\Gamma_{k+1}$.

We now bound $\|A^m(I - A)\|$ uniformly over all linear maps $A\colon \mathbb{C}^n \to \mathbb{C}^n$ that can be expressed as a product of $k$ complex projections by applying the above bound on the numerical range into the result of [CP17] that there exists a constant $Q \le 1 + \sqrt{2}$ such that

$$\|p(A)\| \le Q \sup_{z \in W(A)} |p(z)|$$

for all polynomials $p$. We defer the calculation to the supplementary material.

**Lemma 9.**

$$\sup_{z \in \Gamma_k} |z^m(1 - z)| \le \frac{k}{m}\left(\frac{4}{e\pi^2} + o_{k, m \to \infty}(1)\right).$$

A more precise version of this bound in which $\frac{4}{e\pi^2}$ can be improved in terms of $\frac{k}{m}$ will be given in the supplementary material.

This, combined with Crouzeix's inequality, gives Lemma 2, which we restate below more precisely, and hence Theorem 1.

**Lemma 2.** *For any linear map $A\colon \mathbb{C}^n \to \mathbb{C}^n$ expressible as a product of $k$ complex projections,*

$$\|A^m(I - A)\| \le \frac{k}{m}\left(\frac{4}{e\pi^2} + o_{k, m \to \infty}(1)\right) Q$$

*where $Q$ is the constant of Crouzeix's inequality as presented above.*

## 4   Real projections

In the application to forgetting, all relevant projections are real projections. Therefore one may ask whether our bound can be improved by restricting to products of $k$ real projections.

As we show in the supplementary material, the union of the numerical ranges is the same even if we restrict to real orthogonal projections $\mathbb{R}^4 \to \mathbb{R}^4$, and it is possible to get the maximal asymptotic decay rate on $\|A^m - A^{m+1}\|$, but any real or complex sequence of orthogonal projections attaining a point on $\partial\Gamma_k$ in its numerical range can be orthogonally decomposed into invariant (under all projections) subspaces $U \oplus V$ where $\|A^m\vec{u}\| - \|A^{m+1}\vec{u}\|$ decays quickly on $U$. In particular, any indecomposable such sequence must have quickly decaying forgetting.

## 5   Future Work

It is not clear that this yields a completely optimal bound for forgetting. Is it possible to improve $O(T^2/m)$ to $O(T/m)$ matching the lower bound of [Evr+22]?

Finally one might wonder how rare the worst-case bound is. For example if one visits the $T$ tasks cyclically, but in a random order is it possible to beat the $O(T^2/m)$ bound, perhaps with some weak dependence on the ambient dimension?

## 6  Acknowledgment

The authors would like to thank Itay Evron for interesting discussions about [Evr+22].

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
