# 1 A geometric interpretation of the forgetting bound in [Evr+22]

To show this, from the previous section, $F_S(mT)$ is at most the average distance from $\vec{w}_{mT}$ to $P_1, P_2, \ldots, P_T$, and $\vec{w}_{mT}$ is on $P_T$. The iterates $\vec{w}_{mT+1}, \vec{w}_{mT+2}, \ldots, \vec{w_{mT+(T-1)}}$ pass through the rest of the planes, so the distance from $\vec{w}_{mT}$ to $P_t$ is at most $\|\vec{w}_{mT} - \vec{w}_{mT+t}\|$. The point will be that if $\|\vec{w}_t\|^2$ does not decrease much from $t = mT$ to $t = (m+1)T$, then $\vec{w}_t$ cannot move much along its path from $t = mT$ to $t = (m+1)T$.

Let $d_t := \|\vec{w}_t - \vec{w}_{t+1}\|$. Then $\|\vec{w}_{mT} - \vec{w}_{mT+t}\| \le d_{mT} + d_{mT+1} + \cdots + d_{mT+t-1}$.

As $\vec{w}_{t+1}$ is a projection of $\vec{w}_t$ onto a subspace, $d_t^2 = \|\vec{w}_t\|^2 - \|\vec{w}_{t+1}\|^2$, so $\|\vec{w}_{mT}\|^2 - \|\vec{w}_{(m+1)T}\|^2 = d_{mT}^2 + d_{mT+1}^2 + \cdots + d_{(m+1)T-1}^2$.

$\frac{F_S(mT)}{\|\vec{w}_{mT} - \vec{w}_{(m+1)T}\|^2} \le \frac{\frac{1}{T}\sum_{t=1}^{T-1}\left(\sum_{s=0}^{t} d_{mT+s}\right)^2}{\sum_{t=0}^{T-1} d_{mT+t}^2}$. While we could now optimize this quadratic form precisely, we opt for a simpler bound, deferring a more precise approach to section 3 : $\left(\sum_{s=0}^{t} d_{mT+s}\right)^2 = \|(d_{mT+s})_{s=1}^{t}\|_{L^1}^2 \le t \|(d_{mT+s})_{s=1}^{t}\|_{L^2}^2$ (with equality when all $d_{mT+s}$ are equal).

Our bound becomes

$$\frac{\frac{1}{T}\sum_{t=1}^{T-1}\left(\sum_{s=0}^{t} d_{mT+s}\right)^2}{\sum_{t=0}^{T-1} d_{mT+t}^2} \le \frac{\frac{1}{T}\sum_{t=1}^{T-1} t\sum_{s=0}^{t} d_{mT+s}^2}{\sum_{t=0}^{T-1} d_{mT+t}^2}$$
$$= \frac{\frac{1}{T}\sum_{s=0}^{T}\sum_{t=s}^{T-1} td_{mT+s}^2}{\sum_{t=0}^{T-1} d_{mT+t}^2}$$
$$\le \frac{\frac{1}{T}\sum_{s=0}^{T}\sum_{t=0}^{T-1} td_{mT+s}^2}{\sum_{t=0}^{T-1} d_{mT+t}^2}$$
$$= \frac{T-1}{2}.$$

# 2 A tighter normalization

Suppose we did not normalize $\max_t \|X_t\| = 1$. Then each loss would be scaled by $\|X_t\|$. The argument in the previous section applies directly with an additional factor of $\frac{1}{T}\sum_{t=1}^{T} \|X_t\|$, so we can replace the normalizing of $\max_t \|X_t\| = 1$ with $\sum_{t=1}^{T} \frac{1}{T} \|X_t\| = 1$ without affecting the results.

# 3 Optimizing the quadratic form more precisely

[Evr+22] observed that one could improve the bound on forgetting in terms of $\sup_{A,\vec{u}} \|A^{m-1}\vec{u}\|_2^2 - \|A^m\vec{u}\|_2^2$ by a factor of 2 (compared to the bound above in terms of $\|A^m\vec{u}\|_2^2 - \|A^{m+1}\vec{u}\|_2^2$ (with a different exponent) for the specific $A, \vec{u} = \vec{w}_0$ in our forgetting) by using that $\vec{w}_{mT}$ can be shown to be even closer to the latter half of the planes $P_{T-1}, P_{T-2}, \ldots$ by considering its distance to $\vec{w}_{mT-1}, \vec{w}_{mT-2}, \ldots$ and instead choosing $A$ to be the update map shifted by half the iterates, and choosing $\vec{u}$ to be $\vec{w}_{\frac{T}{2}}$ (up to a floor or ceiling). Applying a similar analysis gives the factor of 2.

But it is possible to improve by more than a factor of 2 by computing the optimizer of the ratio of the quadratic forms: The relevant quantity to bound can now be expressed as $\frac{\vec{v}^T Q \vec{v}}{\vec{v}^T \vec{v}}$ for a block matrix $Q$ split along the middle, so it's enough to optimize a single block which is half the length of the full matrix. This is optimized by computing the largest eigenvalue of $Q$, which can be computed by looking at the characteristic polynomial of its inverse to gain another factor of 2 (asymptotically).

This analysis can be combined with removing the normalization of the $X_t$, in which case $Q$ depends on $\|X_t\|$ but the same procedure would work as long as you can compute the eigenvalues of $Q$.

## 4 Proof of Lemma 4

We start with the following simple fact that we will use below.

**Proposition 11.** *Let $x, y \in \mathbb{C}$ be such that $\{\gamma x + \overline{\gamma}y | \gamma \in \mathbb{C}\}$ consists of real multiples of some fixed complex number. Then $|x| = |y|$.*

*Proof.* If $x + y = 0$ then we are done. Otherwise $\frac{ix-iy}{x+y} \in \mathbb{R}$. So

$$\frac{ix - iy}{x + y} = \frac{-i\overline{x} + i\overline{y}}{\overline{x} + \overline{y}},$$

which rearranges to $|x| = |y|$. $\qquad\square$

Now we present the proof of Lemma 4.

*Proof.* For $\vec{z} \in \mathbb{C}^2$ let $\nabla_{m,\vec{z}}P$ denote the directional derivative of $P$ as the $m$th coordinate is varied in the $\vec{z}$ direction. That is,

$$\nabla_{m,\vec{z}}P(\vec{v}_0, \ldots, \vec{v}_{k-1}) := \lim_{t \to 0} \frac{1}{t} \left( P(\vec{v}_0, \ldots, \vec{v}_m + t\vec{z}, \ldots, \vec{v}_{k-1}) - P(\vec{v}_0, \ldots, \vec{v}_{k-1}) \right).$$

A direct computation shows

$$\nabla_{m,\vec{z}}P(\vec{v}_0, \ldots, \vec{v}_{k-1}) = \left( \frac{\langle \vec{v}_{m-1}, \vec{z} \rangle}{\langle \vec{v}_{m-1}, \vec{v}_m \rangle} + \frac{\langle \vec{z}, \vec{v}_{m+1} \rangle}{\langle \vec{v}_m, \vec{v}_{m+1} \rangle} \right) P(\vec{v}_0, \ldots, \vec{v}_{k-1}).$$

Here, and throughout the rest of the argument, indices are treated mod $k$. The denominators above are nonzero unless $P(\vec{v}_0, \ldots, \vec{v}_{k-1}) = 0$. It will become clear that $0$ is not an outermost boundary point of $\Gamma_k$ for $k \geq 3$, so we may ignore this case.

Each $\vec{v}_m^{\perp}$, along with its scalar multiples, lies in the tangent space of the unit sphere at $\vec{v}_m$. So if $(\vec{v}_0, \ldots, \vec{v}_{k-1})$ is a boundary point, then the directional derivatives $\nabla_{m,\gamma\vec{v}_m^{\perp}}$ must all be parallel as $m$ varies over $\{1, \ldots, k\}$ and $\gamma$ varies over the unit circle. Hence for any fixed $m$,

$$\left\{ \gamma \frac{\langle \vec{v}_{m-1}, \vec{v}_m^{\perp} \rangle}{\langle \vec{v}_{m-1}, \vec{v}_m \rangle} + \overline{\gamma} \frac{\langle \vec{v}_m^{\perp}, \vec{v}_{m+1} \rangle}{\langle \vec{v}_m, \vec{v}_{m+1} \rangle} \middle| \gamma \in \mathbb{C} \right\}$$

consists of real multiples of some fixed complex number. By Proposition 11 along with the Pythagorean theorem,

$$\frac{1 - |\langle \vec{v}_{m-1}, \vec{v}_m \rangle|^2}{|\langle \vec{v}_{m-1}, \vec{v}_m \rangle|^2} = \left| \frac{\langle \vec{v}_{m-1}, \vec{v}_m^{\perp} \rangle}{\langle \vec{v}_{m-1}, \vec{v}_m \rangle} \right|^2 = \left| \frac{\langle \vec{v}_m^{\perp}, \vec{v}_{m+1} \rangle}{\langle \vec{v}_m, \vec{v}_{m+1} \rangle} \right|^2 = \frac{1 - |\langle \vec{v}_m, \vec{v}_{m+1} \rangle|^2}{|\langle \vec{v}_m, \vec{v}_{m+1} \rangle|^2},$$

so $|\langle \vec{v}_{m-1}, \vec{v}_m \rangle| = |\langle \vec{v}_m, \vec{v}_{m+1} \rangle|$ for all $m$.

Scaling each $\vec{v}_m$ by a complex number $\phi_m$ with unit norm does not change the value of $P$. However this scales the inner products $\langle \vec{v}_{m-1}, \vec{v}_m \rangle$ by $\overline{\phi_{m-1}}\phi_m$. By choosing appropriate $\phi_m$'s, we can make $\overline{\phi_{m-1}}\phi_m \langle \vec{v}_{m-1}, \vec{v}_m \rangle$ constant in $m$. To see this, identify a unit complex number with its argument, so that multiplication corresponds to addition mod $2\pi$ and conjugation corresponds to negation. The vectors $\vec{e}_0 - \vec{e}_1, \vec{e}_1 - \vec{e}_2, \ldots, \vec{e}_{k-1} - \vec{e}_0 \in \mathbb{R}^k$ span the set of vectors whose coordinates sum to $0$. So choosing the $\phi_m$ appropriately allows us to make

$$\{\overline{\phi_{m-1}}\phi_m \langle \vec{v}_{m-1}, \vec{v}_m \rangle\}_{m=1}^k$$

any list of complex numbers of norm $\alpha$ with product $P(\vec{v}_0, \ldots, \vec{v}_{k-1})$. In particular they can all be made equal.

Thus any boundary point is achieved by a sequence $(\vec{v}_0, \ldots, \vec{v}_{k-1})$ with $\langle \vec{v}_m, \vec{v}_{m+1} \rangle = \alpha$ for a single $\alpha \in \mathbb{C}$, so for the remainder of this proof, we assume $(\vec{v}_0, \ldots, \vec{v}_{k-1})$ is a critical point with this property.

Set

$$\beta_m = \langle \vec{v}_m, \vec{v}_{m+1}^{\perp} \rangle = -\overline{\langle \vec{v}_m^{\perp}, \vec{v}_{m+1} \rangle},$$

64   so

$$\frac{\alpha}{P(\vec{v}_0, \ldots, \vec{v}_{k-1})} \nabla_{m, \vec{v}_m^{\perp}} P(\vec{v}_0, \ldots, \vec{v}_{k-1}) = (\beta_{m-1} - \overline{\beta_m}).$$

65   Again using that the derivatives are parallel at a critical point,

$$(\beta_0 - \overline{\beta_1}), (\beta_1 - \overline{\beta_2}), \ldots, (\beta_{k-1} - \overline{\beta_0}) \in \{\lambda z | \lambda \in \mathbb{R}\} =: \ell_z$$

66   for some $z \in \mathbb{C}$ of unit norm. By the Pythagorean theorem, $|\beta_m|^2 = 1 - \alpha^2$, so all $\beta_m$'s have the
67   same norm. This implies that either (i) $\beta_{m-1}$ and $-\overline{\beta_m}$ are reflections about $\ell_z$, or (ii) $\beta_{m-1} = \overline{\beta_m}$.
68   If (ii) holds for some $m$, then

$$\frac{\alpha}{P(\vec{v}_0, \ldots, \vec{v}_{k-1})} \nabla_{m, i\vec{v}_m^{\perp}} P(\vec{v}_0, \ldots, \vec{v}_{k-1}) = (i\beta_{m-1} + i\overline{\beta_m}) = 2i\beta_{m-1}.$$

69   So as long as the $\beta$'s are nonzero, $2i\beta_{m-1}$ must be a real multiple of $z$ (because
70   $\frac{\alpha}{P(\vec{v}_0, \ldots, \vec{v}_{k-1})} \nabla_{m, \vec{v}_m^{\perp}} P(\vec{v}_0, \ldots, \vec{v}_{k-1}) = (\beta_{m-1} - \overline{\beta_m})$ all lie on $\ell_z$). This means that $z$ is per-
71   pendicular to $\beta_{m-1}$. When $\beta_{m-1} = \overline{\beta_m}$ this implies that $-\overline{\beta_m}$ is the reflection of $\beta_{m-1}$ over $\ell_z$. So
72   condition (i) continues to hold in this case (and it holds trivially if the $\beta$'s are 0).

73   Thus we may assume that (i) holds for all $m$. Then $\beta_m$ is the image of $\beta_{m-1}$ under a reflection about
74   $\ell_z$ composed with a reflection about the imaginary axis. The composition of these reflections is a
75   rotation about the origin, and hence corresponds to multiplication by some unit norm $\omega \in \mathbb{C}$. Thus
76   $\beta_m = \omega^m \beta_0$ for all $m$, and also $\beta_0 = \beta_k = \omega^k \beta_0$. So either $\omega$ is a (not necessarily primitive) $k$th
77   root of unity or $\beta_0 = \ldots = \beta_{k-1} = 0$. Either way, one can write $\beta_m = \omega^m \beta_0$ where $\omega$ is a $k$th root
78   of unity. $\qquad \square$

## 79   5 Any (possibly inconsistent) cyclic sequence of $T$ tasks converges to a cycle of
## 80     length $T$

81   Recall that the direction of an affine subspace is the vector space spanned by any two vectors in that
82   subspace.

83   The claim is equivalent to the following by setting $w$ to be $w_{i+kT}$ and cycling $P_1, \ldots, P_T$ so that
84   $P_{i\%T+1}$ comes first. (Or alternatively, by setting $w = w_0$ and using that the claim is preserved under
85   applying any affine map.)

86   **Proposition 12.** *Let $P_1, P_2, \ldots, P_T$ be a sequence of affine subspaces of $\mathbb{R}^d$, and let $\vec{w} \in \mathbb{R}^d$.*
87   *Let $M : \mathbb{R}^d \to \mathbb{R}^d$ be the affine map given by the composition of orthogonal projections onto*
88   *$P_1, P_2, \ldots, P_T$ in that order. Then $\vec{w}, M\vec{w}, M^2\vec{w} \ldots$ converges to a fixed point of $M$, either linearly*
89   *or after a finite number of iterations.*

90   *Proof.* We first show that the restriction of $M$ to the affine hull $A$ of $\vec{w}, M\vec{w}, M^2\vec{w} \ldots$ is a strict
91   contraction.

92   Indeed, the only projections that do not decrease $\|M^{k+1}\vec{w} - M^k\vec{w}\|$ are those onto affine subspaces
93   whose direction contains $M^{k+1}\vec{w} - M^k\vec{w}$, but any sequence of such projections sends $M^k\vec{w}$ to points
94   in the affine subspace orthogonal to $A$ through $M^k\vec{w}$, so it is not possible that all the projections
95   $P_i, P_{i+1}, \ldots$ are parallel to this vector or else it would be impossible for their composition to send
96   $M^k\vec{w}$ to $M^{k+1}\vec{w}$).

97   Next, the sequence $\vec{w}, M\vec{w}, M^2\vec{w}$ are the partial sums of $\vec{w} + (M\vec{w} - \vec{w}) + M(M\vec{w} - \vec{w}) + M^2(M\vec{w} -$
98   $\vec{w}) + \ldots$ and $\|M^k(M\vec{w} - \vec{w})\|$ is at most $\|(M\vec{w} - \vec{w})\|$ times the operator norm of the linear part
99   of $M \restriction_A$ to the power $k$. This operator norm is strictly less than 1, as it's a strict contraction, so the
100   series converges linearly or faster.

101   To get the lower bound, apply the same argument to the eventual affine hull (the intersection of all
102   affine hulls of subsequences obtained by ignoring a prefix), on which $M$ must act invertibly. If it's
103   only one point, then it must have converged after a finite number of steps; Otherwise, the invertibility
104   of $M$ implies that the smallest singular value is positive. $\qquad \square$

105 We remark that this convergence implies that the forgetting converges to some positive value along
106 each subsequence $w_i, w_{i+T}, w_{i+2T}, \ldots$, but it doesn't necessary converge for the whole sequence:
107 Consider three lines bounding a right triangle. The cycle will include both the right-angle vertex and
108 a point on the hypotenuse, and the forgetting at the point on the hypotenuse is strictly larger than the
109 forgetting at the vertex.

110 **Proposition 13.** *Using the notation of the previous proposition, let $w_{*,1}$ and $w_{*,2}$ be fixed points*
111 *of $M$. Then $w_{*,2} - w_{*,1}$ is contained in the direction of each $P_i$. In other words, letting $D$ be the*
112 *intersections of the directions for the $P_i$, the fixed points are $w_{*,1} + D$.*

113 *In particular, if $P := \bigcap_i P_i \neq \emptyset$, then all fixed points of $M$ are in $P$.*

114 *Proof.* $\|M(w_{*,2} - w_{*,1})\| = \|w_{*,2} - w_{*,1}\|$, but all projections that do not decrease $\|w_{*,2} - w_{*,1}\|$
115 contain $w_{*,2} - w_{*,1}$ in their direction.

116 In particular, if $P \neq \emptyset$, then any point $w_* \in P$ is a fixed point, so all fixed points of $P$ can be
117 expressed as $w_* + d$ where $d$ is in the direction of all the $P_i$, so $w_* + d \in P$. $\qquad\square$

## 6 Proof of Lemma 8

119 The following lemma gives a bound that depends on $\frac{k}{m}$, from which Lemma 8 will follow.

120 **Lemma 14.** *Define $S : [0, \frac{1}{2}] \to [\frac{1}{4}, \infty)$ by $S(t) := \frac{\cot(\pi t)}{2\pi(1-2t)}$ (which is strictly monotonically*
121 *decreasing). Then*

$$\sup_{z \in \Gamma_k} |z^m(1-z)| \leq \begin{cases} \frac{k}{m} \left( e^{-\frac{m}{k}((1-S^{-1}(\frac{m}{k}))S^{-1}(\frac{m}{k}))(2\pi^2)} 2\sin(\pi S^{-1}(\frac{k}{m})) + o_{k\to\infty}(1) \right) & m \leq 4k \\ \frac{k}{m} \left( 2e^{-\frac{k}{m}\frac{\pi^2}{2}} + o_{k\to\infty}(1) \right) & m > 4k \end{cases}.$$

122 *where the little-o terms are uniform in $m$.*

123 *Proof.* We parameterize the boundary of $\Gamma_{k+1}$ by $((1-t) + te^{\frac{2\pi i}{k}})^k$ as $t$ ranges from 0 to 1; By the
124 maximum principle, $\sup_{z \in \Gamma_{k+1}} |(1-z)z^m|$ is attained for some value of $t$.

125 The first factor $1 - z$ becomes

$$\lim_{k\to\infty} \left| 1 - ((1-t) + te^{\frac{2\pi i}{k}})^k \right| = \left| 1 - \lim_{k\to\infty} ((1-t) + te^{\frac{2\pi i}{k}})^k \right|$$

$$= \left| 1 - \lim_{k\to\infty} (e^{\frac{2\pi it}{k}})^k \right|$$

$$= \left| 1 - e^{2\pi it} \right|$$

$$= 2|\sin(\pi t)|$$

126 uniformly in $t$.

127 The second factor $z^m$ becomes

$$\left|(((1-t)+te^{\frac{2\pi i}{k}})^k)^m\right| = \left|(1-t)+te^{\frac{2\pi i}{k}}\right|^{km}$$

$$= \left(((1-t)+t\cos(\frac{2\pi}{k}))^2 + (t\sin(\frac{2\pi}{k}))^2\right)^{\frac{km}{2}}$$

$$= \left((1-t)^2 + 2(1-t)t\cos(\frac{2\pi}{k}) + t^2\right)^{\frac{km}{2}}$$

$$= \left(((1-t)+t)^2 - 2(1-t)t(1-\cos(\frac{2\pi}{k}))\right)^{\frac{km}{2}}$$

$$= \left(1 - 2(1-t)t(1-\cos(\frac{2\pi}{k}))\right)^{\frac{km}{2}}$$

$$= \left(1 - 2(1-t)t(\frac{\left(\frac{2\pi}{k}\right)^2}{2} + O(k^{-4})))\right)^{\frac{km}{2}}$$

$$= \left(1 - \frac{1}{\left(\frac{k^2}{(2(1-t)t)(2\pi^2 + O(k^{-2}))}\right)}\right)^{\frac{km}{2}}$$

$$= \left(\left(1 - \frac{1}{\left(\frac{k^2}{(2(1-t)t)(2\pi^2 + O(k^{-2}))}\right)}\right)^{\frac{k^2}{(2(1-t)t)(2\pi^2+O(k^{-2}))}}\right)^{\frac{m}{2k}(2(1-t)t)(2\pi^2+O(k^{-2}))}$$

128 where the $O(k^{-2})$ term is $-k^2(\cos(\frac{2\pi}{k}) - 1 + \frac{\left(\frac{2\pi}{k}\right)^2}{2})$ in all occurrences.

129 Letting $\alpha := \frac{m}{k}$, this is
$$\left(e^{-1} - o_{k\to\infty}(1)\right)^{\alpha((1-t)t)(2\pi^2 + O(k^{-2}))}.$$

130 Furthermore, $\left(1 - 2(1-t)t(1-\cos(\frac{2\pi}{k}))\right)^{\frac{k}{2}}$ is increasing in $k$ for $k$ sufficient large $k$.

131 (indeed, the derivative of its log with respect to $k$ is $\frac{2\pi m \cdot (1-t)t\sin\left(\frac{2\pi}{k}\right)}{\left(1-2(1-t)t\cdot\left(1-\cos\left(\frac{2\pi}{k}\right)\right)\right)k} +$

132 $\frac{m\ln\left(1-2(1-t)t\cdot\left(1-\cos\left(\frac{2\pi}{k}\right)\right)\right)}{2}$ which is positive for large $k$ (the first addend is positive and the second

133 is negative for large $k$; The limit of the ratio is -2 uniformly in $t \in [0, \frac{1}{2}]$ as $k \to \infty$ so the first term

134 is larger than the second.)) So the limit is from below. That is, the second factor is

$$\left(e^{-((1-t)t)(2\pi^2)} - o_{k\to\infty}(1)\right)^\alpha$$

135 where the little-o is positive.

136 Putting the two factors together, letting $q(z) = z^{mk}(1-z^k)$,

$$|q(1-t+te^{\frac{2\pi i}{k}})| = \left(e^{-((1-t)t)(2\pi^2)} - o_{k\to\infty}(1)\right)^\alpha 2|\sin(\pi t)| = \left(e^{-((1-t)t)(2\pi^2)} - o_{k\to\infty}(1)\right)^\alpha 2\sin(\pi t)$$

137 uniformly on $t \in [0, 1]$. As the little-o is positive, the limit as $k \to \infty$ is also uniform in $m$.

138 For any fixed $\alpha$, the maximum is attained either at an endpoint or where the derivative with respect to

139 $t$ is 0. By symmetry about $\frac{1}{2}$, it suffices to bound this for $t \in [0, \frac{1}{2}]$.

140 The derivative is
$$2\pi e^{-\alpha((1-t)t)(2\pi^2)}(\cos(\pi t) - 2\pi\alpha(1-2t)\sin(\pi t)),$$

141 which, for $t \in (0, \frac{1}{2})$, has the same sign as

$$\frac{\cot(\pi t)}{2\pi(1-2t)} - \alpha.$$

142 As $\frac{\cot(\pi t)}{2\pi(1-2t)}$ is decreasing in $t$ (its derivative has the same sign as $2\pi t + \sin(2\pi t) - \pi$, so letting $s = 2\pi t$,

143 this is $s + \sin(s) - \pi$ which is increasing in $s$), $e^{-\alpha((1-t)t)(2\pi^2)}2\sin(\pi t)$ is increasing in $t$ from 0 until

144 $\frac{\cot(\pi t)}{2\pi(1-2t)} = \alpha$, and after this it is decreasing in $t$. In particular, if there is no $t$ such that $\frac{\cot(\pi t)}{2\pi(1-2t)} = \alpha$

145 (or equivalently, $\alpha < \min_{[0,\frac{1}{2}]} \frac{\cot(\pi t)}{2\pi(1-2t)} = \frac{1}{4}$), then $\operatorname{argmax}_t e^{-\alpha((1-t)t)(2\pi^2)}2\sin(\pi t) = \frac{1}{2}$. $\qquad\square$

146 The bound from the main paper follows:

**Lemma 8.**

$$\sup_{z \in \Gamma_k} |z^m(1-z)| \leq \frac{k}{m}\left(\frac{4}{e\pi^2} + o_{k,m\to\infty}(1)\right).$$

147 *Proof.* If $m \leq 4k$ then the maximum of the limit as $k, m \to \infty$ with $\alpha$ fixed is attained at a point

148 where $\frac{m}{k} = \alpha = \frac{\cot(\pi t)}{2\pi(1-2t)}$. Plugging this in gives

$$\lim_{k\to\infty, m=\alpha k} |q(1 - t + te^{\frac{2\pi i}{k}})|^{\frac{m}{k}} = e^{-\frac{\cot(\pi t)}{2\pi(1-2t)}((1-t)t)(2\pi^2)}2\sin(\pi t)\frac{\cot(\pi t)}{2\pi(1-2t)},$$

149 which is bounded by

$$\frac{1}{e\pi} \leq e^{-\frac{\cot(\pi t)}{2\pi(1-2t)}((1-t)t)(2\pi^2)}2\sin(\pi t)\frac{\cot(\pi t)}{2\pi(1-2t)} \leq \frac{1}{2e^{\frac{\pi^2}{8}}},$$

150 where equality is attained for the first inequality at $t = 0$ (which corresponds to $\alpha = \infty$) and for the

151 second at $t = \frac{1}{2}$ (which corresponds to $\alpha = \frac{1}{4}$).

152 If $k > \frac{m}{4}$ then the maximum is attained at $t = \frac{1}{2}$, giving the value $\frac{k}{m}\left(2e^{-\frac{k}{m}\frac{\pi^2}{2}}\right)$. $\qquad\square$

# 7 Reducing to $\omega = 1$ via quaternions

154 We rewrite the arguments that reduce our problem to solving the equation when $\omega = 1$ (possibly

155 replacing $k$ with $2k$) purely in terms of quaternions.

156 For $k$ odd, then for any solution to

$$1 = (\alpha + \beta\omega^{k-1}j)(\alpha + \beta\omega^{k-2}j)\dots(\alpha + \beta\omega^0 j),$$

157 it holds that

$$1 = ((\alpha\zeta_k) + (\beta\zeta_k)(\zeta_k^{-2}\omega)^k j)((\alpha\zeta_k) + (\beta\zeta_k)(\zeta_k^{-2}\omega)^{k-1}j)\dots((\alpha\zeta_k) + (\beta\zeta_k)(\zeta_k^{-2}\omega)^0 j).$$

158 For $k$ even, the original proof could be translated directly into quaternions, but we find the following

159 slightly modified version cleaner: For any solution to

$$1 = (\alpha + \beta\omega^{k-1}j)(\alpha + \beta\omega^{k-2}j)\dots(\alpha + \beta\omega^0 j),$$

160 it holds that

$$((\alpha\zeta_{2k}) + (\beta\zeta_{2k})(\zeta_{2k}^{-2}\omega)^k j)((\alpha\zeta_{2k}) + (\beta\zeta_{2k})(\zeta_{2k}^{-2}\omega)^{k-1}j)\dots((\alpha\zeta_{2k}) + (\beta\zeta_{2k})(\zeta_{2k}^{-2}\omega)^0 j) = \zeta_{2k}^k = \pm 1.$$

161 Therefore the square of the left hand side is 1.

162 The whole problem has a simple expression in terms of quaternions: letting $\mathfrak{C}(a+bi+cj+dk) = a+bi$

163 be the complex part and $\mathfrak{H}(a+bi+cj+dk) = c+di$ be the "quaternionic part", the desired range

164 is the same as the range of $\prod_{m=0}^{k-1} \mathfrak{C}p_m$ subject to the constraint $\prod_{m=0}^{k-1} p_m = 1$. (An equivalent

165 definition is $\mathfrak{C}(p) = \frac{p + ipi}{2}$.)

166 Indeed, the inner product $\langle p, q \rangle$ where $p, q \in \mathbb{H}$ is $\mathfrak{C}q^{-1}p$, so letting our sequence of vectors be

167 $p_0, p_0 p_1, p_0 p_1 p_2, \dots, p_0 p_1 \dots p_k$ for $p_m \in \mathbb{H}$ gives the claim.

168 The constraint that all inner products are equal becomes the constraint that $\mathfrak{C}p_m$ is the same for all $m$.

## 7.1 The range of $P$ includes the interior

To show that the range of $P : (\mathbb{C}^2)^k \to \mathbb{C}$ includes the interior, we inductively (in $k$) give a geometric description of the range. For clarity, we let $P_k$ denote the version of $P$ with domain $(\mathbb{C}^2)^k$.

The range of $P_1$ is $\{1\}$.

For any $k \in \mathbb{Z}_{\geq 0}$, any point in the range of $P_{k+1}$ can be obtained by picking a number $p \in \Gamma_k$, taking a sequence $(a_0, b_0), (a_1, b_1), \ldots, (a_{k-1}, b_{k-1}) \in \mathbb{C}^2$ with $P_k((a_0, b_0), (a_1, b_1), \ldots, (a_{k-1}, b_{k-1})) = p$, and adding a point $(a_k, b_k)$ to the end of the sequence.

Fix any sequence $(a_0, b_0), (a_1, b_1), \ldots, (a_{k-1}, b_{k-1}) \in \mathbb{C}^2$. The set of possible ratios $\frac{P_{k+1}((a_m, b_m)_{m=0}^k)}{P_k((a_m, b_m)_{m=0}^{k-1})} = \frac{\langle (a_{k-1}, b_{k-1}), (a_k, b_k) \rangle \langle (a_k, b_k), (a_0, b_0) \rangle}{\langle (a_{k-1}, b_{k-1}), (a_0, b_0) \rangle}$ depends only on $\langle (a_0, b_0), (a_{k-1}, b_{k-1}) \rangle$.

This range is the union of circles centered at each point in $[0, 1]$, where the radii vary like an ellipse, as a rescaled version of $R(x) = l\sqrt{1 - x^2}$, and the center gives the new squared inner product with $(a_0, b_0)$.

By using a unitary transformation sending $(a_2, b_2)$ to $(1, 0)$, every such point can be expressed in 2 ways (up to multiplicity). These two coordinates form the new squared inner product magnitudes.

To show that this set is contractible, we show that its intersection with any line with fixed real part is contractible and show that applying the contraction to each intersection gives a line segment. More generally, we use the following lemma and corollary:

**Lemma 15.** *Let $\{f_i\}_{i \in I_{\leq 0}}, \{f_j\}_{j \in I_{\geq 0}}$ be families of (not necessarily continuous) partial functions $\mathbb{R}^n \to \mathbb{R}$ where $f_i \geq 0, f_j \leq 0$. Assume these families are connected under the norm $\|f - g\| = \sup_{x \in \mathbb{R}^n} |f_0(x) - g_0(x)|$ where $h_0$ denotes the extension of the partial function $h$ to all of $\mathbb{R}^n$ by 0.*

*Let $D_{\leq 0} = \bigcup_{i \in I_{\leq 0}} D(f_i), D_{\geq 0} = \bigcup_{j \in I_{\geq 0}} D(f_j)$ where $D(f)$ denotes the domain of definition of $f$, and let $D = D_{\leq 0} \cup D_{\geq 0}$.*

*Define $f_{\leq 0, \sup} : D_{\leq 0} \to \mathbb{R}$ by $f_{\leq 0, \sup}(x) := \sup_{\substack{i \in I_{\leq 0} \\ x \in D(f_i)}} f(x)$ and $f_{\geq 0, \inf} : D'_{\geq 0} \to \mathbb{R}$ by $f_{\geq 0, \inf}(x) := \inf_{\substack{j \in I_{\geq 0} \\ x \in D(f_j)}} f(x)$.*

*Then the (closed) region bounded by $\Gamma(f_{\leq 0, \sup}) \cup \Gamma(f_{\geq 0, \inf}(x))$ where $\Gamma(f)$ denotes the graph of $f$ in $\mathbb{R}^{n+1}$ (which may have multiple components) can be expressed as*

$$D \times \{0\} \cup \left( \bigcap_{i \in I_{\leq 0}} \{(x, y) : x \in D(f_i), f_i(x) \leq y \leq 0\} \right) \cup \left( \bigcap_{j \in I_{\geq 0}} \{(x, y) : x \in D(f_j), 0 \leq y \leq f_j(x)\} \right).$$

*Let $X = \left( \bigcup_{i \in I_{\leq 0}} \Gamma(f_i) \right) \cup () \bigcup_{j \in I_{\geq 0}} \Gamma(f_j)$*

*If $f_{\leq 0, \sup}$ and $f_{\geq 0, \inf}(x)$ are continuous, then $\overline{X}$ deformation retracts to*

$$G := \left( \bigcup_{x \in D_{\leq 0}} \sup\{f_i(x) : i \in I_{\leq 0}\} \right) \cup \left( \bigcup_{x \in D_{\geq 0}} \inf\{f_j(x) : j \in I_{\geq 0}\} \right).$$

*A deformation retract is given by linearly decreasing the magnitude of the last coordinate.*

*Proof.* The first claim factors over $\mathbb{R}^n$, so it suffices to prove this claim when $D$ is a single point, in which case it is trivial.

For the second claim, by the connectedness of each family, their images on a single value of $x$ are connected with constant sign, so the proposed deformation retract is well-defined. By the continuity of $f_{\leq 0, \sup}$ and $f_{\geq 0, \inf}$, the proposed deformation retract is continuous. $\square$

We leave finding the correct generalization when $f_{\leq 0, \sup}$ and $f_{\geq 0, \inf}(x)$ are discontinuous to the interested reader.

**Corollary 16.** *Let $D \subset \mathbb{R}^n \subset \mathbb{R}^{n+1}$ be closed, where the embedding sends the last coordinate to 0.*
*Let $R : D \to \mathbb{R}_{\geq 0}$.*

*Let $X = \bigcup_{x \in D} S_{R(x)}(x)$ where $S_r(x)$ denotes the sphere of radius $r$ centered at $x$.*

*Letting $P_{n+1} : \mathbb{R}^{n+1} \to \mathbb{R}^n$ be the projection sending the last coordinate to 0, define $G_{\min} : P_{n+1}(X) \to \mathbb{R}_{\geq 0}$ by $G_{\min}(x) := \min\{y \in \mathbb{R}_{\geq 0} : (x, y) \in X\}$, where the minimum exists because $D$ is closed. Similarly define $G_{\max} : P_{n+1}(X) \to \mathbb{R}_{\geq 0}$ be defined by $G_{\max}(x) := \max\{y \in \mathbb{R}_{\geq 0} : (x, y) \in X\}$. Then:*

1. *$X = \bigcup_{x \in P_{n+1}(X)}\{(x, y) : G_{\min}(x) \leq |y| \leq G_{\max}(x)\}$.*

2. *$X$ deformation retracts to $\bigcup_{x \in P_{n+1}(X)}\{(x, y) : G_{\min}(x) = |y|\}$.*

3. *The region bounded by this surface, $\bigcup_{x \in P_{n+1}(X)}\{(x, y) : G_{\min}(x) < |y|\}$, is $\bigcap_{x \in D} B_{R(x)}(x)$,*

*In particular, if $R(x)$ is ever 0, then $X$ deformation retracts to $P_{n+1}(X)$.*

*In this case, if $D$ is convex, then $X$ is contractible.*

*Proof.* Because $D$ is closed, $X = \overline{X}$. Then the lemma gives the 3 enumerated items.

For what remains, it suffices to find a deformation retract from $X$ to $D$. It is given by sending every point in $X \setminus D$ in the direction towards the nearest point in $D$, which is unique because $D$ is convex. $\qquad\square$

One can generalize to higher dimensional spheres (in which case the dimension of the codomain also increases) by working on each copy of $\mathbb{R}^{n+1}$ containing $\mathbb{R}^n$ independently. A similar generalization applies to the lemma.

In our case, we can find the range by doing a calculation on each vertical line. Alternatively, if the radius at $t$ is given by $R(t)$ a (not necessarily strict) superset of the boundary is the union of the circles at each endpoint plus the curve parameterized by $(t, 0) + R(t)(-R'(t), \pm\sqrt{1 - R'(t)^2})$ where $t$ ranges over the line segment (this curve parameterizes, for each $t$ in the interior of the interval where $|R'(t)| < 1$, the unique point accessible from that point and not any nearby points. If $|R'(t)| > 1$, then no such points exist.).

This range forms an ellipse $\frac{x^2}{l^2+1} + \frac{y^2}{l^2} = 1$ with foci at $(0, 0)$ and $(1, 0)$ by working one vertical line at a time.

The condition that all consecutive inner products are equal corresponds to the condition that the extremal points all come from the same point twice in each step of the geometric construction.

One gets another constraint because any two ellipses constructed in the above way passing through a specified point have different directions of tangency at that point, so the inner products $\langle(a_m, b_m), (a_{m+2}, b_{m+2})\rangle$ must also be equal in magnitude.

# 8   Optimizing sequences for $P$ are coplanar

**Proposition 17.** *Any sequence of unit vectors $v_0, v_1, \ldots, v_{k-1} \in \mathbb{C}^n$ with $P(v_0, v_1, \ldots, v_{k-1}) \in \Gamma_k$ must be coplanar (i.e., lie on a complex plane).*

*Proof.* We may assume $k \geq 3$, because any 2 vectors are coplanar.

We prove the contrapositive: If $v_0, v_1, v_2, \ldots, v_{k-1}$ are not coplanar, then $P(v_0, v_1, v_2, \ldots, v_{k-1})$ is in the interior of $\Gamma_k$.

As $\Gamma_k$ is radial and $\partial\Gamma_k$ is continuous (as a function of the complex argument), it suffices to show that there exists a sequence of $k$ vectors whose image under $P$ has the same argument but a larger magnitude.

As 0 is in the interior of $\Gamma_k$, we may assume none of the $v_i$ are 0.

248 Assume without loss of generality that $v_0, v_1, v_2$ are not coplanar. Let $v_1'$ denote the projection of $v_1$
249 onto the plane spanned by $v_0, v_2$. Then

$$\frac{P(v_0, \frac{v_1'}{\|v_1'\|}, v_2, \ldots, v_{k_1}}{P(v_0, v_1, v_2, \ldots, v_{k-1}))} = \frac{1}{\|v_1'\|} \in \mathbb{R}_{>1}.$$

250 □

# 9 Products of projections with numerical range intersecting $\partial\Gamma_k$

252 As shown in the proof of Theorem 5, any sequence of unit vectors $v_0, v_1, \ldots, v_{k-1} \in \mathbb{H}$ realizing
253 $P(v_0, \ldots, v_{k-1}) \in \partial\Gamma_k$ must be obtainable from a sequence of the form $v_0, v_0 u, v_0 u^2, \ldots, v_0 u^{k-1}$
254 (where $u$ is a quaternionic $k$th root of unity) by multiplying each vector by a complex unit.

255 The interpretation in the sense of projections is that, if $v_0$ realizes $\partial\Gamma_k$, then the sequence of
256 projections must send $v_0$ to the sequence of vectors formed by projecting onto $v_0 u, v_0 u^2, \ldots$ in that
257 order (multiplying by a complex unit does not change the line we project onto at each step). Up to a
258 unitary transformation, we may take $v_0 = 1$.

259 Writing $u = a + bi$ with $a, b \in \mathbb{C}$, by direct computation, the sequence of projections is

$$1, au, a^2 u^2, \ldots.$$

260 Combining this with the result from the previous section that any sequence of vectors optimizing
261 $P$ must be coplanar, we get that if $A = P_k P_{k-1} \ldots P_1$ is a project of $k$ projections on $\mathbb{C}^d$ with
262 $W(A) \cap \Gamma_{k+1} \neq \emptyset$, then for any vector $v_0 \in \mathbb{C}^d$ with $v_0^* A v_0 \in \partial\Gamma_k$, the vectors $P_i P_{i-1} \ldots P_1 v_0$
263 must all be coplanar, and furthermore there must exist a unitary transformation $\mathbb{H} \to \mathbb{C}^d$ such that
264 the sequence of projections is the image of $1, \bar{a} u, \bar{a}^2 u^2, \ldots.$

# 10 Real projections

266 We can directly show that any product $A$ of $k$ real projections whose numerical contains a point in
267 $\partial\Gamma_k$ must be decomposable into a direct sum $U \oplus V$ of subspaces, invariant under each projection,
268 such that $\|A^m \vec{u}\|^2 - \|A^{m+1} \vec{u}\|^2$ is small for all $\vec{u}$.

269 **Proposition 18.** *If $P_k, \ldots, P_1 \colon \mathbb{R}^n \to \mathbb{R}^n$ are orthogonal projections satisfying $\gamma \in W(P_k \ldots P_1)$*
270 *for some $\gamma \in \partial\Gamma_k$, then there is an 4-dimensional subspace $V \subseteq \mathbb{R}^n$ invariant under each $P_i$ such*
271 *that $\gamma \in W(P_k \restriction_V \ldots P_1 \restriction_V)$.*

272 *Proof.* As we show in the supplementary material, if $P_k, \ldots, P_1 \colon \mathbb{C}^n \to \mathbb{C}^n$ are complex projections
273 and $v \in \mathbb{C}^n$ such that $\gamma = v^T P_k \ldots P_1 v \in \partial\Gamma_k$, then $v, P_1 v, \ldots, P_k \ldots P_1 v$ must lie in a complex
274 plane. Combining this with the above equality case gives that there must exist an isometry of complex
275 vector spaces $\mathbb{H} \to \mathbb{C}^n$ such that the action of the projections on the image corresponds to an equality
276 case.

277 In particular, if $P_1, \ldots, P_k$ are real projections, then the copy of $\mathbb{R}^4$ spanned by the real parts of
278 the image of $f$ is invariant under all $P_t$, and the numerical range of this restriction also intersects
279 $\partial\Gamma_k$. □

280 To get the orthogonal decomposition, the orthogonal complement of any subspace invariant under
281 all projections is invariant under all projections. The reason is that this is true for each projection
282 individually (i.e., for any orthogonal projection, the orthogonal complement of any invariant subspace
283 is invariant).

284 Taking the invariant subspace from the proposition, we get that there is an invariant subspace where
285 the projections act as described in the previous section. But the norms of these vectors decay
286 geometrically, and therefore cannot do asymptotically better than the lower bound given by [Evr+22],
287 and furthermore any collection of projections that does better can be done without having numerical
288 range of the product intersect $\partial\Gamma_k$ by removing all orthogonal summands of this form.

## 11 Existence of real realizations

Despite the previous section, one may independently wonder whether the bound on the numerical range can be improved by restricting to real projections, thus improving our forgetting bound. The answer is that it cannot: Real projections can have product with a numerical range including any point of $\partial\Gamma_k$.

We will show that, for any unit quaternion $u = \alpha + \beta j \in \mathbb{H}$ (with $\alpha, \beta \in \mathbb{C}$), there exists a sequence of vectors $\mathbf{u}_{n,\Re}, \mathbf{u}_{n,\Im} \in \mathbb{C}^4$, such that:

1. The unitary map of complex vector spaces $\Phi : \mathbb{H} \to \mathbb{C}^4$ sending $u^0 = 1 \mapsto \mathbf{u}_{0,\Re} + i\mathbf{u}_{0,\Im}$ and $u^1 \mapsto \mathbf{u}_{1,\Re} + i\mathbf{u}_{1,\Im}$ sends $u^n \mapsto \mathbf{u}_{n,\Re} + i\mathbf{u}_{n,\Im}$. (If $\beta = 0$ then $\Phi$ is not determined by the $\Phi(u^0)$ and $\Phi(u^1)$, but $\Phi(u^n)$ always is.)

2. The real projection onto $\text{Span}(\mathbf{u}_{n,\Re}, \mathbf{u}_{n,\Im})$ (complexified to a map $\mathbb{C}^4 \to \mathbb{C}^4$) sends $\Phi(\overline{\alpha}^{n-1}u^{n-1})$ to $\Phi(\overline{\alpha}^n u^n)$ (equivalently, it sends the real (resp. imaginary) part to the real (resp. imaginary) part).

The second condition is equivalent to saying that the real projection onto $\text{Span}(\mathbf{u}_{n,\Re}, \mathbf{u}_{n,\Im})$ sends $\Phi(u^{n-1})$ to $\Phi(\overline{\alpha}u^n)$.

Let

$$\mathbf{u}_{0,\Re} + i\mathbf{u}_{0,\Im} = \frac{1}{\sqrt{2}}\left(\begin{pmatrix} 1 \\ 0 \\ 0 \\ 0 \end{pmatrix} + i\begin{pmatrix} 0 \\ 1 \\ 0 \\ 0 \end{pmatrix}\right)$$

and

$$\overline{\alpha}\left(\mathbf{u}_{1,\Re} + i\mathbf{u}_{1,\Im}\right) = \frac{1}{\sqrt{2}}\left(\begin{pmatrix} |\alpha|^2 \\ 0 \\ \sqrt{|\alpha|^2 - |\alpha|^4} \\ 0 \end{pmatrix} + i\begin{pmatrix} 0 \\ |\alpha|^2 \\ 0 \\ \sqrt{|\alpha|^2 - |\alpha|^4} \end{pmatrix}\right).$$

As all powers of $u$ are real linear combinations of $1$ and $u$ (via the recurrence relation $u^2 = -1 + (2\Re u)u$, which holds for any unit quaternion), the first property above determines $\mathbf{u}_{n,\Re}, \mathbf{u}_{n,\Im}$.

We next check that there exists a unitary map $\Phi$ is unitary. Indeed, all we need to check is that

$$\alpha = \langle \mathbf{u}_{0,\Re} + i\mathbf{u}_{0,\Im}, \mathbf{u}_{1,\Re} + i\mathbf{u}_{1,\Im}\rangle,$$

or equivalently

$$|\alpha|^2 = \overline{\alpha}\alpha = \langle \mathbf{u}_{0,\Re} + i\mathbf{u}_{0,\Im}, \overline{\alpha}(\mathbf{u}_{1,\Re} + i\mathbf{u}_{1,\Im})\rangle$$

which is true.

Finally, we check the second property. This is invariant under (real) rotations.

**Definition 19.** *Two unit vectors $\mathbf{1}, \mathbf{u} \in \mathbb{C}^4$ are **compatible** with respect to $\alpha \in \mathbb{C}$ if both of the following hold:*

- *The complex projection of $\mathbf{1}$ onto $\mathbf{u}$ is $\overline{\alpha}\mathbf{u}$, and is a real projection*

- *The complex projection of $\overline{\alpha}\mathbf{u}$ onto $\mathbf{u}^2$ is $\overline{\alpha}^2\mathbf{u}^2$, and is a real projection.*

Two vectors being compatible means that $\mathbf{1}, \mathbf{u}, \mathbf{u}^2$ (with the last defined by the recurrence relation) doesn't violate the second condition (though $\mathbf{u}^3, \ldots$ might).

The property of two vectors being compatible is invariant under rotations of $\mathbb{R}^4$ and multiplication by any complex unit (where both vectors must be multiplied by the same complex unit).

As before, in the second condition we may replace $\overline{\alpha}\mathbf{u}$ with $\mathbf{u}$, $\mathbf{u}^2$ with any complex multiple of $\mathbf{u}^2$, and $\overline{\alpha}^2\mathbf{u}^2$ with $\overline{\alpha}\mathbf{u}^2$.

Write $\mathbf{1} = \overline{\alpha}\mathbf{u} + \mathbf{V}$, so

$$\Re\mathbf{V}, \Im\mathbf{V} \perp \Re\mathbf{u}, \Im\mathbf{u}$$

 (not respectively: All four pairs are orthogonal. Indeed, because the complex projection of $\mathbf{1}$ onto $\overline{\alpha}\mathbf{u}$ is a real projection, the real and imaginary parts of $\mathbf{V}$ must be orthogonal to the plane spanned by $\Re\mathbf{u}, \Im\mathbf{u}$.) Then

$$\mathbf{u^2} = -\mathbf{1} + 2\Re\alpha\mathbf{u} = (2\Re\alpha - \overline{\alpha})\mathbf{u} - \mathbf{V} = \alpha\mathbf{u} - \mathbf{V}$$

$$\overline{\alpha}\mathbf{u^2} = |\alpha|^2\mathbf{u} - \overline{\alpha}\mathbf{V}.$$

So the second condition in the definition of compatibility is equivalent to the complex projection of $\mathbf{u}$ onto $|\alpha|^2\mathbf{u} - \overline{\alpha}\mathbf{V}$ being $|\alpha|^2\mathbf{u} - \overline{\alpha}\mathbf{V}$ and being a real projection.

So the condition is that there is a real projection sending $\Re\mathbf{u}$ to $|\alpha|^2\Re\mathbf{u} - \Re(\overline{\alpha}\mathbf{V})$ and $\Im\mathbf{u}$ to $|\alpha|^2\Im\mathbf{u} - \Im(\overline{\alpha}\mathbf{V})$.

As $\Re\mathbf{V}, \Im\mathbf{V} \perp \Re\mathbf{u}, \Im\mathbf{u}$, there exists such a real projection if and only if both of the following hold:

- $\Re(\overline{\alpha}\mathbf{V}) \perp \Im(\overline{\alpha}\mathbf{V})$, or equivalently $\Re\mathbf{V} \perp \Im\mathbf{V}$. This is true by construction.
- $|\Re(\overline{\alpha}\mathbf{V})|^2 = 1 - |\alpha|^2|\Re\mathbf{u}|^2$ and $|\Im(\overline{\alpha}\mathbf{V})|^2 = 1 - |\alpha|^2|\Im\mathbf{u}|^2$ .

(Indeed, being of the right length means there exists a real projection sending $\Re\mathbf{u}$ to $\Re(\overline{\alpha}\mathbf{u^2})$; Conditional on this, the possible projections of $\Im\mathbf{u}$ are the sphere with a diameter formed by its projection onto the projection of $\Re\mathbf{u}$ (which is 0 because everything is perpendicular) and its projection onto the orthogonal complement of $\Re u$.)

A sequence of vectors works if and only if all consecutive pairs except the last two are compatible with $\overline{\alpha}$. For this, we need the real and imaginary components of everything to be orthogonal, but we multiply by $\overline{\alpha}$ each time so the only way being orthogonal like this is preserved is if $\overline{\alpha}$ is purely real or purely imaginary (which only corresponds to a nontrivial point on the boundary if $k = 2$) or if $\|\Re\mathbf{u}\| = \|\Im\mathbf{u}\|$. The latter case uniquely determines the projections.

The constructed realization also shows that the asymptotic supremum of $\|A^m - A^{m+1}\|$ is the same for real projections as for complex projections.

As an aside, this also implies that this can't be obtained using projections onto subspaces of codimension 1, because the real and imaginary parts of the $\mathbf{u^n}$ have to be orthogonal, and that can't be preserved under taking a projection onto a subspace of codimension 1 unless one of the vectors is in the subspace.

## 12 Remark on the task dependency on forgetting

One might also be interested in exploring the question of how task dependency affects forgetting in general. One way to capture the task dependency is through the Friedrichs number or its like as they govern the geometric decay rate of residual errors and forgetting for a fixed set of tasks. Specifically, if you consider any sequence of $T$ number of fixed tasks, their Friedrichs angle (for $T = 2$) and its extension, the Friedrichs number (for $T > 2$), are always less than 1 [AS16; BS16]. This causes the residual error to converge geometrically [BS16]. As a result, the rate of forgetting also converges geometrically, and not inversely proportional to the number of iterations, as suggested by our bounds or by Evrons' [Evr+22] for worst-case scenarios. We refer readers interested in such results to [BS16].

## 13 Forgetting vs. Regret

In our context (assuming consistent tasks), one could define the regret for a sequence of tasks $S$ at iteration $n$ as

$$R_S(n) := \frac{1}{n}\sum_{t=1}^{n}\|X_t w_t - y_t\|^2,$$

in contrast to the forgetting which was defined as

$$F_S(n) := \frac{1}{n}\sum_{t=1}^{n}\|X_t w_n - y_t\|^2.$$

While superficially similar, analyzing regret is quite different (and much simpler) than analyzing forgetting in our setting. Indeed the regret over the first $k$ iterations is simply the sum of squares of the update distances. By iterating the Pythagorean Theorem, one can see

$$\sum_{t=1}^{n} \|X_t w_t - y_t\|^2 = \|w_n\|^2 + \sum_{t=1}^{n} \|w_t - w_{t-1}\|^2 = \|w_n - w_0\|^2 \leq 4 \|w_0\|^2$$

since $w_t - w_{t-1}$ is orthogonal to $w_t$ for $t \geq 1$. This means $R_S(n) \leq O(1/n)$, which is tight even if convergence occurs after a single iteration. (Meaning that $w_1$ satisfies all constraints.)