# OpenReview forum: "Nearly Optimal Bounds for Cyclic Forgetting"
_NeurIPS.cc/2023/Conference — NeurIPS 2023 poster_

### Official Review · Reviewer_kHo7 · 2023-06-20

**Soundness:** 3 good
**Presentation:** 3 good
**Contribution:** 3 good
**Rating:** 6
**Confidence:** 2

**Summary:**

Authors provide theoretical bounds on the forgetting quantity in the continual learning setting for linear tasks, where each round of learning corresponds to projecting onto a linear subspace.
For a cyclic task ordering on $T$ tasks and an arbitrary iteration $m$, they prove the upper bound of $O(T^3/m) $ on the forgetting. For the proof they use a bound on the numerical range of a product of T projections.

**Strengths:**

-near optimal bounds are proven for the worst case for cyclic task ordering(also improving the bounds for less general settings if I understood the related work section correctly): their upper bound $O(T^2/m)$ compared to a $\Omega(T/m)$ lower bound and the previous known upper bounds $O(T^2/\sqrt{mT})$.

-also provides a bound on the numerical range of a product of $T$ projections

**Weaknesses:**

(Common) notations could be defined earlier or there could be a reference to the noation section when first notaions are used in the introduction.

There is still a gap of T if I understood everything correctly.

As I am unfamiliar with some pieces of related work and Math/other details were not carefully checked so it is hard to rate the novelty.

**Questions:**

What is the differnce between the $T^3/m$ bound in the abstract and the $T^2/m$ bound presented in the paper?

**Limitations:**

As they mention in the conclusion there are some gaps for real projections and  for forgetting, all relevant projections are real projections.

---

> ### Author Rebuttal · Authors · 2023-08-10
>
> Notation: We will clarify the notation in the introduction as suggested.
>
> Gap of $T$: To clarify: The upper bound given in our paper has worse $T$-dependence than the lower bound given in [Evr+22], but it has the same $T$-dependence as their upper bound for any fixed dimension. Our bound of $O(\frac{T^2}{m})$ may appear to have worse $T$-dependence than their dimension-independent bound of $\frac{T^2}{\sqrt{mT}}$ if $m\ll T$, but in this setting both our bounds are worse than the trivial bound of 1.
>
>
> Real projections: We have since resolved the real case in the following sense: [Evr+22] bounds the forgetting after $m$ cycles of $k$ tasks in terms of $\sup_A\lVert A^m(I-A)\rVert$ where $A$ ranges over all products of $k$ (real) orthogonal projections. Our paper bounds $\sup_A\lVert A^m(I-A)\rVert$ in terms of $\sup_{z\in W(A)}\lvert z^m(1-z)\rvert$ where $A$ ranges over all products of $k$ orthogonal projections and $W(A)$ denotes the numerical range of $A$. Our original paper computed the supremum over all products of complex orthogonal projections, leaving open whether adding the condition that the projections are real improves the bound.  We have since shown that real projections can attain the same supremum,
>
> Abstract typo: The $T^3/m$ in the abstract is a typo and should say $T^2/m$.

---

> > ### Comment · Area_Chair_2N3e · 2023-08-18
> >
> > To the authors: your response has been read and is being considered.

---

### Official Review · Reviewer_SAj1 · 2023-07-05

**Soundness:** 3 good
**Presentation:** 3 good
**Contribution:** 2 fair
**Rating:** 5
**Confidence:** 3

**Summary:**


The paper studies the setting of continual learning for linear tasks, and in particular the phenomena of catastrophic forgetting.
They prove the best known upper bound on forgetting for the setting of cyclic tasks.

**Strengths:**

The paper demonstrates an upper bound on Forgetting that is independent of the dimension.
This is a strong improvement for settings with very large dimension spaces.


**Weaknesses:**


I think the main weakness is the significance of contribution, compared to previous work.
This result build on recent work by [Evron et al.22] and demonstrates improvement in a certain setting (cyclic).
The paper would be strengthened if either the results would be extended to other setting, and incorporated experimental evaluations - both of which similarly to [Evron et al.22].

Additional comments:

1. Not clear why the paper use the term “near-optimal” :
- The known lower bound from [Evron et al.22] at T^2/m. The bound given in this work has a worse dependence on T compared to [Evron et al.22]. The improvement comes for large dimensions, which is indeed important but not necessarily near-optimal ?

2. Forgetting vs. Regret:    by Evron et al., in the cyclic setting both quantities of Forgetting and Regret will go to 0 as m increases. Since regret is a well studied quantity, it is interesting to compare these and examine how will the convergance rate behave ? a connection between them may allow to build on the vast literature on regret.


Minor comments:
- Notation: I assume you mean $X_i \in \mathbb{C}^{r_i \times d}$. Maybe say r_i refers to samples each with d dimensions.
- Line 92 : “We give nearly a nearly optimal bound”
- Citations do not indicate the names of all authors, incorrect format
- Lines 52-53 - why write T^2/*mT instead of T/m ?

**Questions:**

See above.

**Limitations:**

none.

---

> ### Author Rebuttal · Authors · 2023-08-10
>
> Significance: We would be happy to add a remark about other settings and compare our approach to that of [Evron et al.22]. The challenge with experimental results is that our bounds are worst-case bounds. Empirically, nearly-worst-case examples seem to be quite infrequent.
> However we could include a simple experiment showing empirical performance on the examples given in [Evron et al.22] that were used to obtain (theoretical) lower bounds on forgetting.
>
> Near-optimal bounds: Please see the response to reviewer kHo7 above titled "Gap in $T$." In addition, we would be happy to modify the title to replace "Nearly Optimal" with "Dimension Independent" or "Improved ... for high dimensions", since we also agree that the improvement comes for high dimensions, which is quite important.
>
> Forgetting vs. Regret: We think that regret is also interesting to study. A worst-case bound on the regret is indeed possible, however is fairly simple and does not require our results. To give a quick sketch: In our setting one can bound the regret (see Evron et. al. for a definition) by the average of the squared increments $\|w_{k+1} - w_k\|^2.$  By iterating the Pythagorean theorem one sees that $1 = \|w_0\| = \|w_k - w_*\|^2 + \|w_k - w_{k-1}\|^2 + \ldots + \|w_1 - w_0\|^2$ which gives a (tight) bound of $1/k$ on the regret.  If one defines regret with the non-squared loss, then it turns out that one can give a tight bound of $1/\sqrt{k}$ using a slightly more complicated (but still elementary) argument. (Both bounds are attainable in the cyclic setting, and both bounds also hold without the cyclic assumption.)
>
> Taking average regret over the last cycle is more relevant to our results. In this case, with $m$ cycles of $T$ tasks each, the average regret becomes $\frac{\|w_{mT}\|^2-\|w_{m(T-1)}\|^2}{T}=\frac{\Delta_{m-1}(w^*)}{T}$. Combining this with the equation in line 98 of our paper bounds the worst-case forgetting by $\frac{T(T-1)}{2}$ times the worst-case average regret over the last cycle.
>
> Minor comments: We will correct the formatting and notation as suggested.  Our original reason for writing $\frac{T^2}{mT}$ was to make it look more similar to the other bounds in that sentence ($\frac{T^2}{\sqrt{mT}}$ and $\frac{T^2d}{mT}$), and thus easier to compare; An earlier version of our paper stated our main bound as $\frac{T^3}{mT}$. Furthermore, $mT$ is the total number of iterations, which may help interpret the bound. However, we ended up stating our main result using $\frac{T^2}{m}$ because, as an independent theorem, the first advantage does not apply.

---

> > ### Comment · Area_Chair_2N3e · 2023-08-21
> >
> > To the authors: your response has been read and is being considered.

---

> > ### Comment · Reviewer_SAj1 · 2023-08-22
> >
> > Thank you for your answer, and for the suggested modifications.

---

### Official Review · Reviewer_eHkq · 2023-07-09

**Soundness:** 3 good
**Presentation:** 2 fair
**Contribution:** 2 fair
**Rating:** 3
**Confidence:** 3

**Summary:**

This paper describes bounds for cyclic forgetting when an overparametrized linear model is fit successively to a series of tasks. The exact same setting has been studied fairly recently before in the , and the main contribution here is to improve dimension dependence of the bounds.

**Strengths:**

The theory of Theorem 5 is clean and beautiful, and certainly of mathematical interest in its lack of dependence on ambient dimension. The core idea is creative and the writing is good.

Lines 56-62 of the text are valuable for intuition as to why the suboptimal dimension dependence arises, and how to fix it. This is a very interesting insight which could potentially be exploited for more data-instance-dependent bounds in the future.

**Weaknesses:**

Catastrophic forgetting arises especially when there are certain types of dependencies/correlations between the tasks. This entire manuscript is about studying that situation theoretically, but it completely neglects to provide. The manuscript relies on the publication [Evr+22] to set precedent for the problems studied. However, that paper was pathbreaking in setting up the problem and establishing that cyclic task orderings don't suffer catastrophic forgetting, and included much more interpretation and exposition linking the theory result to the motivation for studying the problem. The current manuscript does not give any of this. Instead, the main result obviates all dependencies on the tasks other than their number, so it does not lend much further insight into the problem.

The proofs are far too long and technical for the main body of the paper - they should be moved to the appendices. The entire discussion leading up to Theorem 5 is full of mathematical beauty arising from symmetry - but none of this is in the service of the task of learning from different tasks, only using the generic fact that the tasks are projections.

**Questions:**

- Please make a pass for key typos. The abstract says T^3 instead of T^2; lines 52-53 shouldn't have T in the denominator anywhere; and many more.
- Is there scope to extend these results to parametrize dependencies between tasks?

**Limitations:**

Yes.

---

> ### Author Rebuttal · Authors · 2023-08-10
>
> Precedent for Problem: There are indeed motivations for our work discussed in referenced paper [Evr+22], that we list here, and will elaborate on in our introduction.
>
> Many data sets in machine learning are cyclic or periodic in nature, for example, due to the "day of the week effect" in  financial data or search engine data. In a manufacturing facility using robots, a machine or robot is typically instructed to repeat a series of tasks for producing certain products. Learning this type of task on arrival can be formulated as cyclic continual learning, which is our main subject of study in terms of catastrophic forgetting.
>
> Additionally the methods of cyclic alternating projections (by Von Neumann and Halperin) and cyclic Kaczmarz methods are well-studied methods for solving linear systems. Our work can be thought of as studying the worst-case forgetting of these popular methods.  Equivalently one can think of this as studying residual bounds for (cyclic) Kaczmarz-type algorithms. While very natural, this is somewhat of a new take on analyzing the convergence of these methods.   [Evr+22] mentions this connection, but leaves open the problem of obtaining tight convergence bounds.  We think this problem is sufficiently natural to justify study.
>
> Finally since high-dimensional data is so ubiquitous in machine learning, we believe that the dimension dependence was a major weakness in the bounds of [Evr+22].  Indeed our bound captures a qualitative phenomenon: the worst-case forgetting need not scale at all with the dimension of the ambient data.
>
> The equations in lines 98 and 99 bound the forgetting of any single sequence of $T$ tasks after $m$ cycles by $(1+\sqrt2)\frac{T-1}{2}\sup_z\lvert (1-z)z^m\rvert$ where $z$ ranges over the numerical range of the product of the projections in that sequence. Our paper bounds this over the class of all sequences of $T$ tasks by characterizing the union of their corresponding numerical ranges, but we do not yet have any simple subclass of sequences of tasks over which the supremum of this value over that class has a better bound. We have later characterized all tasks that attain the maximum possible value in each dimension and shown that none of them can optimize the forgetting.
>
> Proofs Location: As a characterization of the union of the numerical ranges of the products of $k$ complex (and, after submission, also real) projections (for any fixed dimension) is mathematically interesting in its own right, we feel that some of the proof should perhaps be left in the main body. However, we agree that it would make sense to move more technical portions to the appendix, leaving more room for discussion.
>
> Typos: We will indeed make a thorough pass for typos. Note that lines 52-53 do not contain a typo, see ``Minor comments" in the Reviewer SAj1 response. We agree that this could be more clear however.
>
> Task Dependencies: Dependencies between tasks is an interesting setting, and we do believe that this is interesting future work. Since this is a relevant question for a reader, we will add a remark discussing the challenges.

---

> > ### Comment · Area_Chair_2N3e · 2023-08-21
> >
> > To the authors: your response has been read and is being considered.

---

### Official Review · Reviewer_U3GB · 2023-07-20

**Soundness:** 3 good
**Presentation:** 2 fair
**Contribution:** 3 good
**Rating:** 7
**Confidence:** 3

**Summary:**

Consider an overparametrized system solving a periodic sequence of tasks in linear (least squares) regression in the following way: starting from a weight vector $w_t$ for task $t$, perform gradient descent  to solve task $t+1$, which, due to overparametrization, will eventually return an exact solution $w_{t+1}$ for task $t+1$. Here $w_0 = 0$ and each task is specified by a set of input vectors $X$ and output labels $y$.

The procedure amounts to projecting $w_t$ onto the solution (hyper-)space of task $t+1$ and implies a loss of information about task $t$ and all previous tasks. This raises the question just how much is forgotten after n tasks have been visited, which can be quantified as the error of $w_n$ averaged over all previous datasets. While this "forgetting" depends on the data, it is still possible to give a worst case bound for the worst possible periodic task sequence with period $T$.

Previous work gives a bound of $T^2/\sqrt{nT}$ and a dimension dependent bound, as well as a lower bound of order $T^2/n$, the paper at hand gives order $T^3/n$, which has optimal dependence on n, though not on T.

This result is obtained by a reduction of the problem to a bound on the numerical range of a polynomial in $T$ projection operators, which is obtained by an intricate analysis, which (because of time constraints) I didn't completely verify, although I didn't find any flaws.

**Strengths:**

The paper approaches an interesting problem with an equally interesting mathematical technique. The given bound is a relevant improvement and to me seems an important contribution to the subject of continual learning.

The problem is clearly stated, the minor limitations of the main result are nicely exposed already in the title by the word "nearly".

**Weaknesses:**

The paper is technically very heavy in the proofs of Lemma 4 and Theorem 5. If possible the authors might sketch the basic ideas of the proof of Theorem 5 in a short paragraph.

The statement of Lemma 4 is somewhat opaque and its relevance to the problem is not immediately clear. It doesn't help that $\Gamma_k$ shows up in the proof of Lemma 4 (l154), while its definition comes on the next page in the statement of Theorem 5.

The paper leans strongly on the reference [Evr+22], without which it is very difficult to understand. At least Theorem 11 in [Evr+22] could be reproduced.

**Questions:**

The various bounds are sometimes expressed in terms of the number of iterations and sometimes in terms of the number of cycles. This may cause some confusion. Would it be possible to unify this?

The $m$ in l 43 and in (1) seems to be a typo.

**Limitations:**

The authors have adequately addressed the limitations of the paper.

---

> ### Author Rebuttal · Authors · 2023-08-10
>
> Technical Proofs: As Reviewer eHkq suggested, we will move part of the proofs to the appendix and include a high level proof in the main body. We summarize the proof of Theorem 5 as follows: To compute the range of $P$, we show that the outer boundary of $P$ is the claimed sinusoidal spiral and show that the range is simply connected, which implies that the range is the filled sinusoidal spiral. Once we get the outer boundary, the simple connectedness follows from a topological argument given in the supplementary material. To compute the outer boundary, as $P$ takes a sequence of vectors to the cyclic product of pairwise inner products, if any of the input vectors is not coplanar with its neighbors, then by projecting it onto the plane spanned by its neighbors and extending it to be unit norm, we can increase the magnitude of the corresponding factors without changing any directions (shown algebraically in the supplementary material), so it suffices to consider the case when all vectors are coplanar, so it suffices to consider $\mathbb{C}^2$. In this case, $P$ is a (real-)smooth map the (real) manifold $(S^1)^n\subset\mathbb{C}^n$ to the (real) manifold $\mathbb{C}$, so any input that gets sent to a boundary point of $P$ has singular Jacobian. That is, the directional derivative in all directions tangent to the domain must be parallel. It turns out that these algebraic conditions can be algebraically manipulated to characterize all critical points and critical values of $P$, and the computation is made simpler by using quaternions.
>
> Reference to $\Gamma_k$: We will remove the reference to $\Gamma_k$ in the proof of Lemma 4 in the main paper by changing the statement to exclude 0 (where the statement is false, but this doesn't affect the rest of the paper).
>
> Reliance on Reference:  We will add a re-statement of Theorem 11 from [Evr+22] and some exposition around this result.  In addition, in light of our response to Reviewer eHkq titled ``Precedent for Problem'', we plan to also add more exposition and motivation, alleviating the heavy leaning on the [Evr+22] reference.

---

> > ### Comment · Reviewer_U3GB · 2023-08-16
> >
> > Thank you for your clarification and the proposed modifications.

---

### Author Rebuttal · Authors · 2023-08-10

We thank the reviewers for their feedback. The following are our responses to the questions and concerns raised by reviewers.

We will move the proofs of the main lemmas and theorems (through Theorem 4) to an appendix.  This will leave room for additional discussion of the connection to previous forgetting bounds, as well as a less technical exposition of our techniques.

Please see the individual responses for more details.

---

### Decision · Program_Chairs · 2023-09-21

**Decision:**

Accept (poster)

**Comment:**

This paper makes a significant advance on a problem that has attracted substantial interest, using sophisticated techniques.